# A single-cell transcriptomic atlas characterizes the silk-producing organ in the silkworm

Yan Ma [1,3], Wenhui Zeng [1,3], Yongbing Ba [2,3], Qin Luo [1], Yao Ou [1], Rongpeng Liu [1], Jingwen Ma [1], Yiyun Tang [1], Jie Hu[1], Haomiao Wang [1], Xuan Tang[2], Yuanyuan Mu [1], Qingjun Li [1], Yuqin Chen [1], Yiting Ran [1], Zhonghuai Xiang[1] & Hanfu Xu [1✉]

The silk gland of the domesticated silkworm *Bombyx mori*, is a remarkable organ that produces vast amounts of silk with exceptional properties. Little is known about which silk gland cells execute silk protein synthesis and its precise spatiotemporal control. Here, we use single-cell RNA sequencing to build a comprehensive cell atlas of the silkworm silk gland, consisting of 14,972 high-quality cells representing 10 distinct cell types, in three early developmental stages. We annotate all 10 cell types and determine their distributions in each region of the silk gland. Additionally, we decode the developmental trajectory and gene expression status of silk gland cells. Finally, we discover marker genes involved in the regulation of silk gland development and silk protein synthesis. Altogether, this work reveals the heterogeneity of silkworm silk gland cells and their gene expression dynamics, affording a deeper understanding of silk-producing organs at the single-cell level.

[1] State Key Laboratory of Silkworm Genome Biology, College of Sericulture, Textile and Biomass Sciences, Southwest University, Chongqing 400715, China.
[2] Shanghai OE Biotech. Co., Ltd., Shanghai 201212, China. [3]These authors contributed equally: Yan Ma, Wenhui Zeng, Yongbing Ba. ✉email: xuhf@swu.edu.cn

Silk is one of the most extraordinary protein materials in nature and is produced by animals with specialized silk-producing organs. At least 140,000 species, including 113,000 insects and 30,000 spiders, have been shown to secrete silk proteins for purposes such as nest construction, defense, and hunting[1–3]. Most attractively, natural silks (e.g., silkworm silk, spider silk and bagworm silk) have unique properties that are not surpassed by artificial or synthetic textile fibers[4–6], including impressive mechanical properties, biocompatibility, and biodegradability, and they have attracted great interest as green biomaterials. Nevertheless, most silk-producing animals, except domesticated silkworms (*Bombyx mori*) cannot be used to produce silk at a commercial scale due to challenges associated with breeding or minuscule silk yields[7,8]. In addition, how silk is precisely synthesized by organs and how silk properties are determined at the molecular level remain obscure. Answering these questions is of great significance to allow humans to better understand and utilize natural silk.

*B. mori* is an economically important insect known for its production of vast amounts of silk. Silkworm silk has been used commercially in textile production for centuries and in biomedical sutures for decades[9]. The silk gland (SG) of *B. mori* is a distinctive organ that synthesizes and secretes the proteins fibroin and sericin, two major components of cocoon silk. The SG is a paired structure that is often divided into three physiologically distinct regions: the anterior silk gland (ASG, consisting of $350 \times 2$ cells), the middle silk gland (MSG, consisting of $255 \times 2$ cells), and the posterior silk gland (PSG, consisting of $520 \times 2$ cells)[10]. Silk fibroin, which is synthesized in the PSG, is transported to the MSG, where it is covered by sericin and then transported to the ASG, the silk protein-processing channel that turns the silk protein complex into liquid silk fiber and spins the silk[11]. Notably, mitotic division of SG cells ceases after stage 25 (6 days post-egg laying)[12], indicating that the SG has completed morphogenesis at this stage. After this time point, the organ size of the SG increases only through an increase in cell volume caused by chromosomal endoreduplication[13]. Surprisingly, in addition to spinning vast amounts of cocoon silk at the final larval instar, *B. mori* also produces a small amount of silk with excellent mechanical properties when newly hatched (day 1 of the first instar, 1L1D) and as a premolting larva[14]. As an organ with remarkable characteristics, the SG of *B. mori* has emerged as a valuable research model for uncovering the mysteries of silk-producing organs, such as how the SG develops, how silk protein is synthesized, and how silk properties are determined. Although previous studies have facilitated our understanding of the SG, little is known about the molecular underpinnings of SG cells, such as the cell types in the SG and their biological functions, cell states at different developmental stages, and gene expression profiles at the single-cell level. Hence, it is important to prospectively characterize the molecular signatures of the SG at single-cell resolution.

In this work, we used single-cell RNA sequencing (scRNA-seq)[15] to detail the development of the *B. mori* SG at early stages (Fig. 1), including the 8 days post-egg laying (E8D) stage that represents the embryonic stage after the complete formation of silk gland; the 1L1D, the first day of larvae, at which the silkworm secretes silk with excellent mechanical properties; and the first larval molting (1LM), a period in which silkworm larvae are undergoing the first organ renewal. By sequencing the cells dissociated from the SG at three representative stages, we built a comprehensive cell atlas of the *B. mori* SG with 14,972 single-cell transcriptional profiles that define 10 distinct cell types. We identified cell-type-specific marker genes, annotated the cell type distributions in each region of the SG, and decoded the developmental trajectories and gene-switch status of the SG cells, which revealed the diversity of SG cells and provided

insights into the regulation of SG development and silk protein synthesis. This single-cell atlas of the *B. mori* SG provides a valuable resource for better understanding the molecular features of silk-producing organs at the single-cell level.

## Results

**Cell type composition of the *B. mori* SG**. *B. mori* developed at E8D, 1L1D, and 1LM were selected for SG sampling. Collecting SGs from these early developmental stages was very challenging. Nevertheless, we finally harvested ~2000 intact SGs for each sample, thereby ensuring the success of subsequent cell dissociation and scRNA-seq analysis using the 10× Genomics platform. After strict quality control filtering, a total of 14,972 high-quality cells were used for subsequent cell clustering and annotation. We aligned cells from E8D, 1L1D, and 1LM and clustered them using the *FindClusters* function in Seurat to catalog the cell types in the *B. mori* SG[16]. Overall, 10 distinct clusters ("cluster" is abbreviated as C hereafter) were classified and visualized using the uniform manifold approximation and projection (UMAP) algorithm (Fig. 2a). The cell number, proportion and genes detected in each cell type were counted. Interestingly, C1, 3, 4, and 6 were dominant in E8D (accounting for 49.08%, 14.27%, 9.9% and 23.07%, respectively); C1, 3, 4, and 9 were dominant in 1L1D (35.34%, 24.37%, 26.62%, and 8.37%, respectively); and C1, 2, 3, 5, 7, and 8 were dominant in 1LM (11.38%, 34.85%, 11.38%, 20.98%, 10.76%, and 7.75%, respectively) (Fig. 2b). The active genes identified in cells of each cluster, referred to as marker genes, are listed in Supplementary Data 1. The top 5 highly expressed markers (Supplementary Data 2) and cluster-specific markers (Supplementary Data 3) of each cluster, which include several previously reported genes such as *P25* and *Btl*, are shown in Fig. 2c, d. Altogether, these results revealed the cell type composition of the *B. mori* SG and reflected the diversity of SG cells.

**Identification of cell types in the SG**. We aggregated all cell clusters according to the expression of markers using the following schemes to identify the cell type in each region of the SG: 1) classic markers were used to annotate cell types by Featureplots and real-time quantitative reverse transcription PCR (qRT–PCR); 2) cluster-specific markers were verified using qRT–PCR and immunofluorescence staining, and 3) GO and KEGG analyses of marker genes in each cell type were performed.

*Cell types in the ASG*. Specifically, 4 of 10 clusters, C4, C5, C8, and C9, mapped to the ASG based on the regional expression of two classic ASG-specific genes *BmASSCP2* and *BGIBMGA011721*[17,18] (C4 and C9) and four cluster-specific markers *LOC101746180* (C5), *LOC101740197* (C8), *Btl* (C9), and *LOC101743237* (C4 and C9) (Fig. 3a; Supplementary Fig. 1a), which were further supported by the immunofluorescence staining for LOC101746180 (C5) and LOC101740197 (C8) (Fig. 3b; Supplementary Fig. 2). Further analyses showed that genes corresponding to "ATP hydrolysis coupled proton transport", "Oxidative phosphorylation" and "mTOR signaling pathway" were enriched in C4, suggesting that this cell type functions in providing basic energy and nutrients for SG development, which is an energy-consuming process driven by ATP[18,19]. C5 and C8, both of which were dominant in 1LM, were enriched in the terms "Polytene chromosome" and "Polytene chromosome puff" and "Chitin metabolic process" and "Chitin binding", respectively, suggesting that these cells are undergoing endoreduplication and may play essential roles in providing lubrication and good permeability in the lumen for silk spinning, as described previously[12,20]. C9 was dominant in 1L1D and enriched in genes corresponding to "Serine-type endopeptidase inhibitor activity" (Supplementary

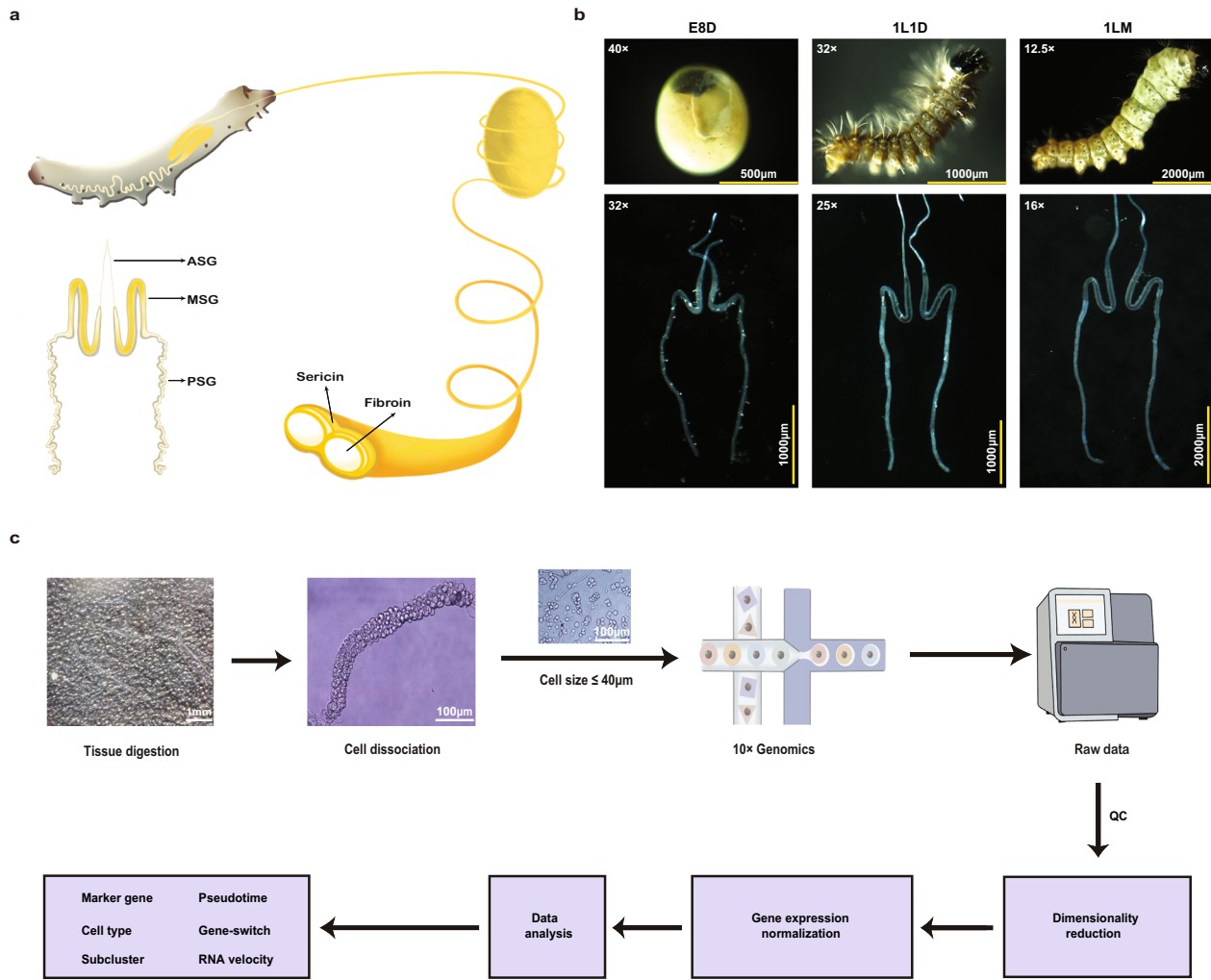

**Fig. 1 Workflow of the scRNA-seq analysis of silkworm silk glands. a** Schematic diagram of the silkworm larva, silk gland, and cocoon silk. The anterior silk gland (ASG) is a processing lumen of liquid silk protein, the middle silk gland (MSG) is responsible for producing silk sericin protein, and the posterior silk gland (PSG) is responsible for producing silk fibroin protein. **b** Silk glands at three developmental stages used for the scRNA-seq analysis. Images show the developing embryo and larvae and corresponding silk glands. E8D, 8 days post-egg laying; 1L1D, day 1 of the first instar; 1LM, the first larval molting. **c** Schematic diagram of sample preparation and workflow of data analysis.

Fig. 3a, b; Supplementary Data 4), indicating that this cell type may function to prevent the degradation of the silk protein before secretion and maintain silk quality and homeostasis during the formation and spinning of silk fibers, as described in previous studies[21,22]. A subcluster analysis of C9 showed that serpin genes (i.e., *serpin 3*, *serpin 5*, *serpin 10*, and *serpin 32*) were enriched in Subcluster 2 while Subcluster 1 mainly included some cuticular proteins (Supplementary Fig. 4a; Supplementary Data 4). Based on these analyses, C4, C5, C8, and C9 are defined as liquid silk fibrosis cells (LSFs), epithelial cell remodeling cells (ECRs), chitin metabolism cells (CMs), and traction forces cells (TRs), respectively (Fig. 3c).

*Cell types in the MSG.* Specifically, 3 of 10 clusters, C3, C6, and C10, mapped to the MSG because they expressed the sericin protein-coding genes *Ser1*, *Ser2*, and *Ser3*[23], and cluster-specific markers *LOC101745308* (C3), *LOC101740733* (C6), and *LOC101744718* (C10), which were further supported by the Featureplots of *Ser2* (C10) and immunofluorescence staining for LOC101745308 (C3) (Fig. 3a, b; Supplementary Figs. 1b, 2). Further analyses showed that genes in C3 were mainly involved in "Translation", "Cytoplasmic translation" and "Translational

initiation", suggesting that this cell type functions in sericin protein synthesis. The E8D-specific cluster C6 was enriched in genes corresponding to "Endoplasmic reticulum lumen", "Pentose and glucuronate interconversions", "Protein processing in endoplasmic reticulum", and "Oxidative phosphorylation", suggesting that this cell type plays an important role in energy metabolism in the SG when the silkworm is about to hatch. C10 was dominant in 1LM and enriched in genes related to "Ubiquitin-dependent protein catabolic process", "Circadian rhythm" and "Longevity regulating pathway – multiple species", indicating that it may play a special role in the characteristic physiological functions of MSG at larval molting (Supplementary Fig. 3c, d; Supplementary Data 4). In addition, a subcluster analysis of C6 showed that *LOC693027* (trehalase gene) is an interesting marker of Subcluster 2 and may be an important target for revealing trehalose metabolism in the SG at the embryonic stage (Supplementary Fig. 4b, Supplementary Data 4). Another interesting marker was the *Ser2* gene identified in Subcluster 2 of C10, suggesting that *Ser2* may function at the molting stage (Supplementary Fig. 4c, Supplementary Data 4). Collectively, according to these analyses, C3, C6, and C10 are defined as sericin protein-synthesizing cells (SPSs), endoplasmic reticulum stress signaling

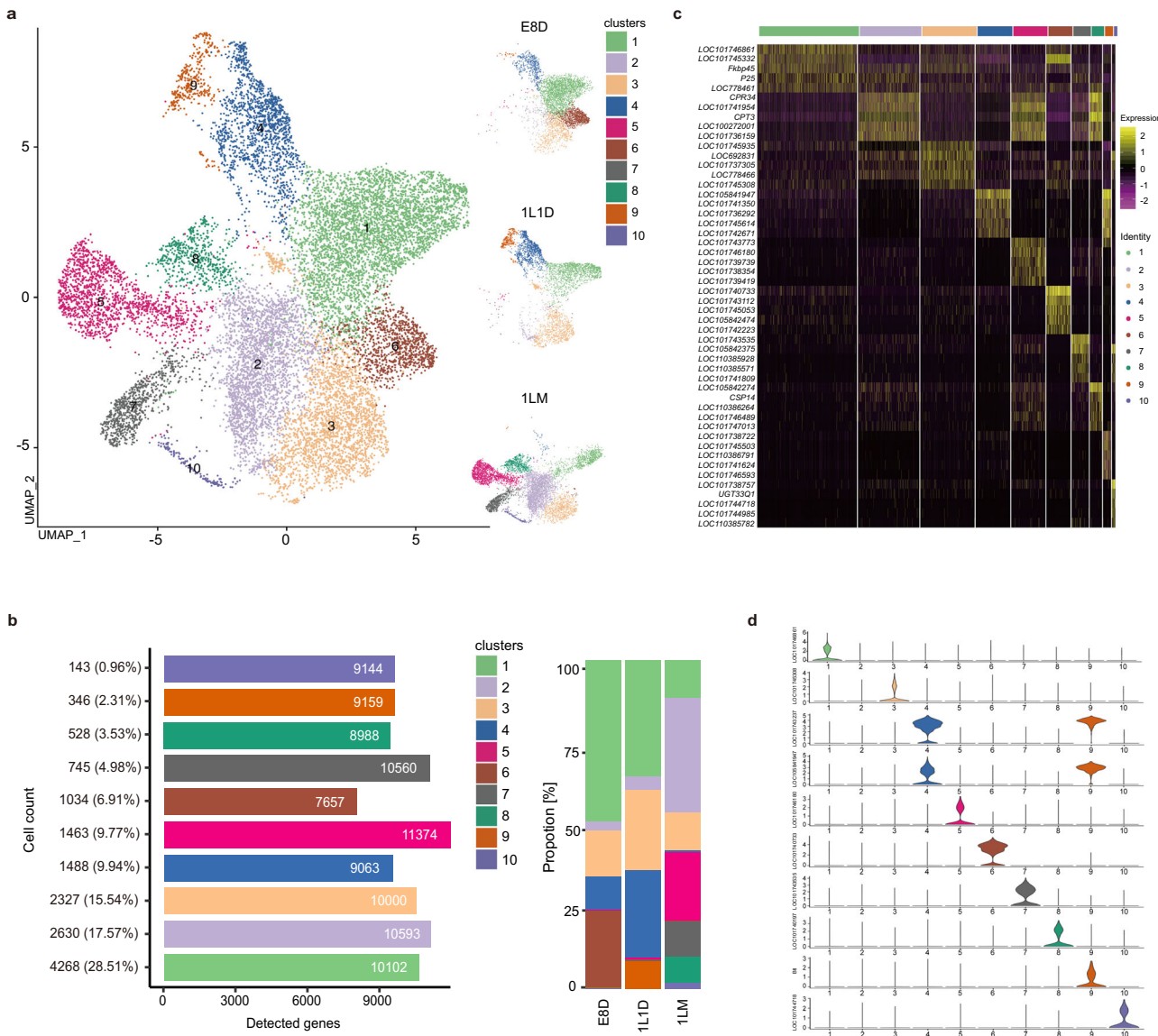

**Fig. 2 Major cell clusters identified in silkworm silk glands. a** UMAP visualization of 10 clusters derived from 14,972 high-quality cells filtered from three silk gland samples. Each dot denotes a single cell. E8D, 8 days post-egg laying; 1L1D, day 1 of the first instar; 1LM, first larval molting. **b** Cell number, proportion and genes detected in each cell cluster. E8D, 8 days post-egg laying; 1L1D, day 1 of the first instar; 1LM, first larval molting. **c** Heatmap showing the expression of top 5 marker genes in each cell cluster. **d** Violin plots showing the expression of representative marker genes in each cell cluster.

cells (ERSSs), and sericin protein catabolism cells (SPCs), respectively (Fig. 3c).

*Cell types in the PSG.* Specifically, 3 of 10 clusters, namely, C1, C2, and C7, mapped to the PSG because they abundantly expressed the fibroin protein-coding genes *fibH*, *fibL*, and *P25*[24], and cluster-specific markers *LOC101746861* (C1) and *LOC101743535* (C7), which were further supported by the results from Featureplots and immunofluorescence staining (Fig. 3a, b; Supplementary Figs. 1c, 2). Further analyses showed that C1 was enriched in genes involved in "Translation" and "Ribosome", indicating that this cell type performs an important function in the translation and transport of silk fibroin. C2 was enriched in genes responding to "Antigen processing and presentation", "B cell receptor signaling pathway" and "TNF signaling pathway", suggesting that this cell type is related to the antibacterial properties of silk. The 1LM-specific cluster C7 was enriched in genes corresponding to "Cellular response to nutrient", "Polytene chromosome puff", "Gene silencing by miRNA" and "Regulation

of autophagy", suggesting that this cell type may perform catabolism-related functions in the PSG at the larval molting stage (Supplementary Fig. 3e, f; Supplementary Data 4). Moreover, C2 was divided into two subclusters, and the fibroin genes *fibH*, *fibL* and *P25* were all located in Subcluster 2, indicating that this sub-cell type mainly performs the function of fibroin protein synthesis (Supplementary Fig. 4d; Supplementary Data 4). In summary, according to these analyses, C1, C2, and C7 are defined as fibroin protein-synthesizing cells (FPSs), death and remodeling regulation cells (DRRs), and fibroin protein catabolism cells (FPCs), respectively (Fig. 3c).

**Early developmental trajectories of SG cells.** We performed pseudotemporal ordering of cells (pseudotime analysis) using Monocle2 to obtain additional insights into the gene expression signatures during SG development[25]. Intriguingly, the pseudotime path of each region in the SG had only one branch, and cell types in the ASG, MSG, and PSG were arranged clearly along the

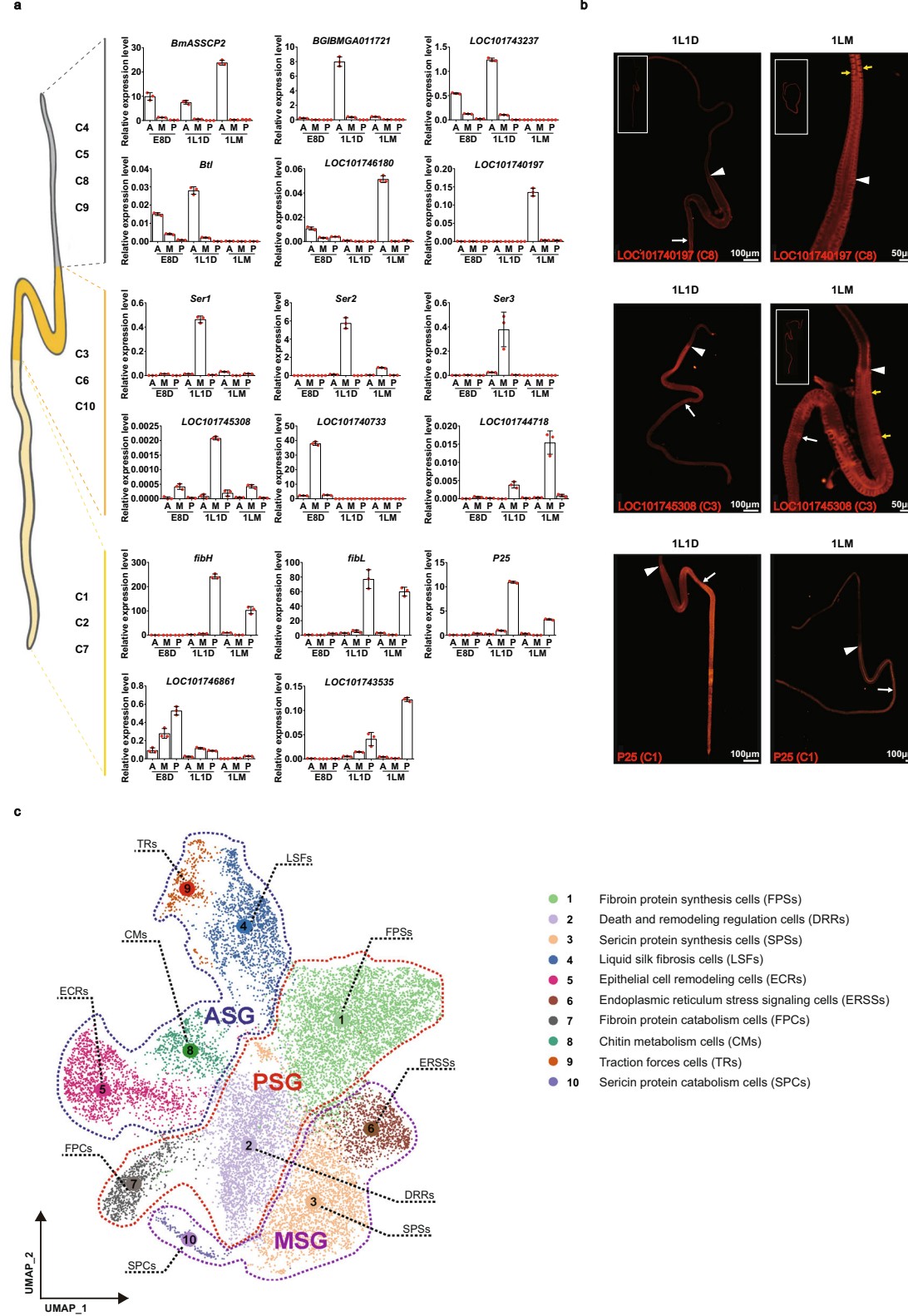

pseudotime path, indicating consistency in the early developmental trend of the three regions of the SG (Fig. 4; Supplementary Fig. 5). Moreover, we analyzed the expression of representative markers (top 10 markers of each cell type) over pseudotime.

Among the cell types of ASG, LSFs and TRs appeared at E8D, increased at 1L1D, and disappeared at 1LM, consistent with their roles in liquid silk protein processing, because the silkworm larvae will spin a small amount of silk protein at 1L1D. ECRs and CMs, both of which function as epithelial cell remodeling cells and chitin metabolism cells, became the major protagonists at 1LM, but ECRs seemed to persist longer than CMs (Fig. 4a; Supplementary Fig. 5). The RNA velocity map showed that TRs developed from LSFs, while ECRs developed from CMs, due to the high RNA velocity (longer arrows) began between them (Fig. 4b). The pseudotime heatmap

**Fig. 3 Verification of cell types in each region of the silk gland. a** qRT–PCR detection of representative marker genes in cells from each region of the silk gland. Data are presented as mean ± SD ($n = 3$ biologically independent experiments). Source data are provided as a Source Data file. A, anterior silk gland; M, middle silk gland; P, posterior silk gland. E8D, 8 days post-egg laying; 1L1D, day 1 of the first instar; 1LM, first larval molting. **b** Immunofluorescence analysis of representative marker genes in each region of the silk gland. The spatial expression pattern of marker genes was shown in red; the yellow arrow indicates the location of protein in silk gland cells. The white arrowhead points to the boundary between the anterior silk gland and middle silk gland; the white arrow points to the boundary between the middle silk gland and posterior silk gland. $n \geq 10$ in three independent experiments. 1L1D, day 1 of the first instar; 1LM, first larval molting. **c** Visualization of 10 cell types using UMAP. Colored dotted lines indicate the cell types present in the anterior silk gland (ASG), middle silk gland (MSG), and posterior silk gland (PSG), respectively.

showed that Modules 1 and 2 were enriched in cuticular protein genes (*CPG6*, *CPH4*, and *CPH19*), dynein-related genes (*LOC101738722* and *LOC105841947*), and an organic cation transporter protein gene (*LOC101745503*), suggesting that these genes may contribute to the process of liquid silk protein conversion in the ASG. Module 3 was characterized by enzyme-coding genes, such as *LOC101742565*, *LOC101746180*, and *LOC105842186*, which may provide an energy supply for cellular metabolism (Fig. 4c; Supplementary Data 5). Collectively, these results provide insights into the developmental trajectory of ASG cells during cell state transitions.

Among the cell types of MSG, ERSSs that are closely related to the nutritional supply of SG before hatching appeared at E8D and disappeared at 1L1D. SPCs appeared at 1LM. SPSs were widely distributed in all three developmental stages, suggesting that the development of this cell type is more complex (Fig. 4a; Supplementary Fig. 5). The RNA velocity of these cell types was consistent with the pseudotime path—larger RNA velocities occur between ERSSs and SPSs or SPSs and SPCs than themselves; thus, SPSs developed from ERSSs and then became SPCs, supporting the pseudotime development of cell types in the MSG (Fig. 4b). The pseudotime heatmap showed that only several genes that appeared in three modules have clear functions or functional annotations, such as *CPH21*, *UGT33R2*, and *Y-d*, but the roles of other genes remain to be verified (Fig. 4c; Supplementary Data 5). Interestingly, eight lncRNAs were detected in three modules, especially in Modules 1 and 3, and we speculate that they may play important roles in the regulation of sericin protein synthesis and metabolic processes.

Regarding cell types in the PSG, FPSs appeared at E8D, were maintained at 1L1D, and gradually decreased at 1LM, suggesting the complex and crucial role of this cell type in fibroin protein synthesis. DRRs increased gradually with progressing developmental stages and FPCs appeared only at 1LM, but the DRRs seemed to perform more functions during development (Fig. 4a; Supplementary Fig. 5). The RNA velocity of these cell types was consistent with the pseudotime path. Briefly, FPSs were segregated and a small part of them transformed to DRRs (short arrows), while the others transformed to FPCs (long arrows). DRRs completely transformed to FPCs, which reflected the rapid activation of FPCs (Fig. 4b). The pseudotime heatmap indicated that most of the top 10 markers in three modules were protein-coding genes (Fig. 4c; Supplementary Data 5). In particular, Modules 1 and 2 included representative genes encoding transport protein (*LOC733070*), cytoplasmic actin (*LOC101741610*), and transmembrane protein (*LOC101737070*), suggesting that they play important roles in the process of silk fibroin protein synthesis and secretion. Module 3 contained four lncRNAs and two zinc-finger transcription factors (TFs) that may be crucial for the regulation of cell fate in the PSG. Collectively, these results improve our understanding of the developmental trajectory of PSG cells.

**Dynamic changes in gene expression in SG cells.** The *B. mori* SG, especially the MSG and PSG, is characterized by its excellent ability to synthesize silk protein, a process that is strictly and cooperatively controlled mainly via the transcriptional response of a large number of genes. We conducted a gene-switch analysis at single-cell resolution based on the genes detected in all clusters to understand how the order (dynamic changes) of gene expression is maintained in different SG cell types (Supplementary Data 6–8). The best-fitting genes, including the top 15 TFs, were plotted along the pseudotime line (Fig. 5; Supplementary Data 9).

In cell types of the ASG, 199 genes were inactivated and most of them were early genes. Remarkably, 2,211 genes were activated, including 12 early genes and 2,045 late genes (Fig. 5a; Supplementary Data 6). Some early and mid genes that may play crucial roles in ASG cells were identified. For example, *Jra*[26], *ABLIM1*[27] and *obst-E*[28] may regulate epithelial morphogenesis, the establishment and maintenance of the cellular structure, and cellular chitin metabolism. *Tret1*[29] may catabolize macromolecular substances and provide energy. Among the late genes, interestingly, we detected numerous TFs, including the 20E response factors Br-c[30], E74[31], and Hr38[32] that have been indicated to be crucial for the regulation of larval molting, and the Awh[33], dimm[34], skd[35], and hth[36] that have been shown in *Drosophila* and *B. mori* to participate in regulating cell function (Fig. 5b; Supplementary Fig. 6a; Supplementary Data 9, 10). These results suggest that early and mid genes that are activated in ASG cells are necessary for maintaining the basic biological functions of the ASG, while a large number of late genes may be responsible for larval molting and metamorphosis, during which numerous TFs are needed to coordinately regulate their expression.

In cell types of the MSG, 47 genes were inactivated. Remarkably, 6329 genes were activated, including 32 early genes, 1487 mid genes, and 4810 late genes (Fig. 5c; Supplementary Data 7). Ribosomal proteins are essential components of the protein biosynthetic machinery. We detected 12 ribosomal protein genes (i.e., *RpS10*, *RpS15*, and *RpL38*) that were activated at an early stage, suggesting active protein synthesis in MSG cells at this stage, which requires the regulation of functional ribosomes assembled by ribosomal proteins. Strikingly, the *fibL*, *fibH*, and *P25* genes were activated at early or mid stages. Additionally, *Ser1*, *Ser2*, and two key TFs, SGF1[37] that directly regulates *Ser1* expression and Awh[33] that directly regulates *fibH* expression, were also detected at midstage. We speculate that the silk protein genes are likely activated in early and mid stages to prepare for larval molting, because the larvae will secrete a small amount of silk proteins in the pre-molting period to steady their bodies. Among late activated genes, the most impressive is the detection of receptors and response factors involved in 20E signaling, including EcR, USP, Hr4, Hr39, Hr96, Br-c, E74, E75, and Ftz-f1, stressing the crucial role of 20E signaling in larval molting. Related to this finding, we also detected 13 cuticular protein genes that were activated in the late stage, which may be necessary for the renewal of larval epidermal tissues (Fig. 5d; Supplementary Fig. 6b; Supplementary Data 9, 10). Collectively, these findings are meaningful for revealing the dynamic changes in gene expression in MSG cells.

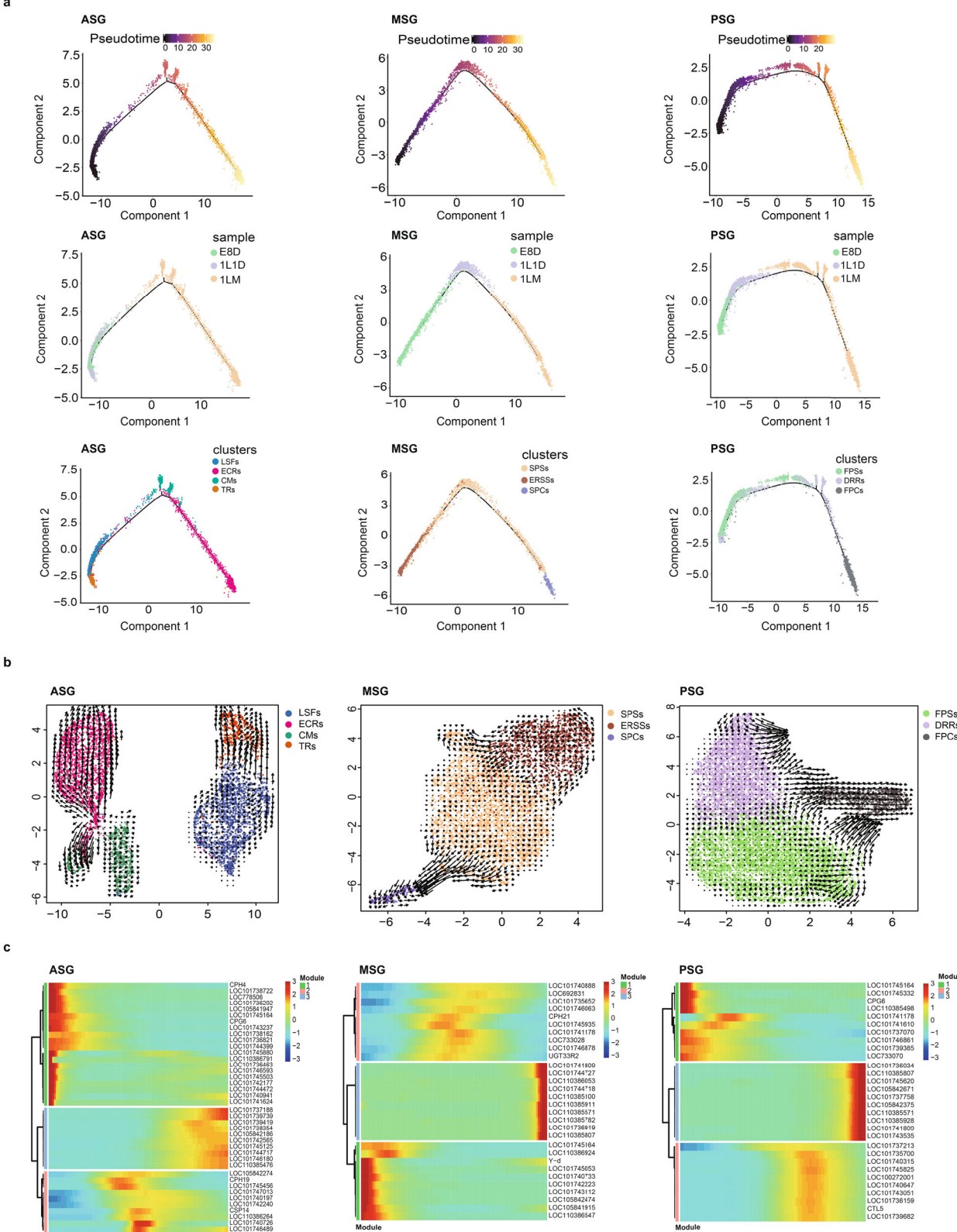

**Fig. 4 Pseudotime analysis of silk gland cells using Monocle2. a** Trajectories of ASG, MSG, and PSG cells along pseudotime. The colors from dark (purple) to light (yellow) represent the forward order of pseudotime. **b** RNA velocity map of ASG, MSG, and PSG cells. The length of the arrows indicates the rate of cell development, the direction of the arrows indicates the direction of cell development. **c** Pseudotime heatmap of the top 10 marker genes (on the right of heatmap) of each cluster in the ASG, MSG, and PSG. ASG anterior silk gland, MSG middle silk gland, PSG posterior silk gland.

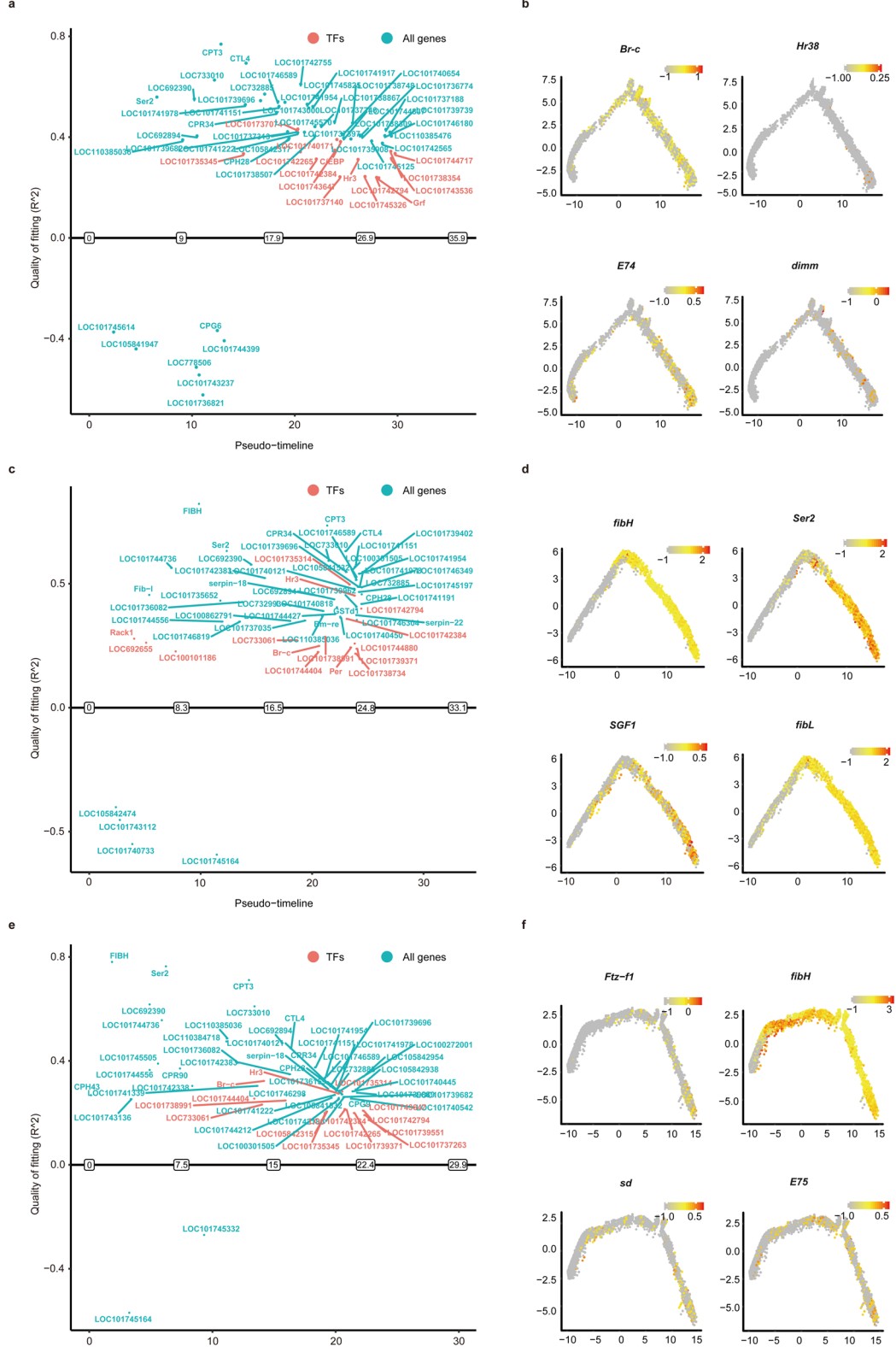

**Fig. 5 Schematic diagram of priority activated and inactivated genes in cell types of the silk gland. a, c, e** Representative activated (top panel) and inactivated (bottom panel) genes in anterior silk gland (**a**), middle silk gland (**c**), and posterior silk gland (**e**) cells. The X-axis shows the predicted open (upper panel) and closed (lower panel) times. The Y-axis represents the goodness of fit. Early genes, switch-at-time < 9.0 (**a**)/8.3 (**c**)/7.5 (**e**); late genes, switch-at-time > 26.9 (**a**)/24.8 (**c**)/22.4 (**e**). TFs, transcription factors. **b, d, f** Trajectory along the pseudotime progression of representative genes that were activated and inactivated in anterior silk gland (**b**), middle silk gland (**d**), and posterior silk gland (**f**) cells. The X-axis represents Component 1, the Y-axis represents Component 2. The formula of this trajectory plot is $\log_{10}$ (value + 0.1).

In cell types of the PSG, 52 genes were inactivated and 4,572 genes were activated (Fig. 5e; Supplementary Data 8). Among the activated genes, *fibH* was activated in the early stage and *P25* was detected in the late stage. Notably, we detected *Ser2* in the early stage and *Ser1* in the late stage. Combined with the findings observed in MSG cells, the result that MSG cells express fibroin genes in addition to sericin genes, while PSG cells express sericin genes in addition to fibroin genes is interesting, which implied that the type of silk protein spun by early larvae is functionally unique and thereby must deploy both the MSG and PSG cells for synthesis. Furthermore, we identified numerous TFs that were expressed at different stages especially in the late stage. For example, SGF1 was detected in the midstage, and the SGF3, dimm, sage, and Awh were activated in the late stage, all of which are known as key TFs regulating silk protein synthesis. Remarkably, 20E signaling was activated in the late stage, including its receptors EcR, USP and response factors E74, E75, Hr4, Hr96, and Ftz-f1. An accumulation of core members of another important pathway, the Hippo signaling pathway (i.e., ex, sav, wts, yki, and sd), was observed and this pathway controls organ size by regulating cell proliferation, apoptosis, and stem cell self-renewal[38]. Surprisingly, 41 lncRNAs were expressed in the early and midstages, but significantly, 244 lncRNAs were expressed in the late stage, implying the essential role for lncRNAs in activating gene expression, especially during larval molting (Fig. 5f; Supplementary Fig. 6c; Supplementary Data 9, 10). Altogether, these results reveal the complex gene expression dynamics in PSG cells, and the many insights into gene activation provided here deserve further study.

**Effective strategies for inducing SG cell growth**. A typical feature of the *B. mori* SG is that once organ morphogenesis is complete by stage 25[12], the cells undergo only endoreduplication but not mitosis, thereby leading to an increase in cell volume. However, the mechanism regulating SG cell growth during early development remains unclear. We found that the volumes of the ASG, MSG, and PSG cells, as well as the intracellular DNA content increased significantly from E8D to 1LM (Fig. 6a, b). Additionally, the space between SG cells was narrower and the connection was tighter in the 1LM stage (Fig. 6a). These observations imply that SG development is closely related to the regulation of the cell number, cell size, and intracellular material circulation and communication. Therefore, we decided to investigate the marker genes that may be involved in regulating these processes during early development.

*Balance of cell number*. Among the marker genes of SG cells, interestingly, numerous genes related to the regulation of cell numbers were identified (Supplementary Data 11). For example, *Btl*, a key gene known for controlling tubular organogenesis, cell proliferation and migration[39–41], was located on the membrane and was expressed at high levels in the ASG and anterior MSG at E8D and 1L1D (Figs. 3a; 6c). *sage*, a key TF that has recently been shown to control cell number in the SG[42], was expressed in SPSs and ERSSs of the MSG and DRRs and FPCs of the PSG. In addition, genes associated with endoreduplication and negative regulation of cell proliferation were also ubiquitously expressed in SG cells, such as *LOC101744399,* which was abundantly expressed in LSFs and TRs, which is a CDK activity-related gene[43] and may play important roles in regulating intracellular replication and inhibiting proliferation. Moreover, genes that may negatively regulate mitotic cell cycle G2/M transition to inhibit cell proliferation, including *PINX1*[44] and *Foxn3*[45], were detected in DRRs, ECRs, FPCs, and SPCs of 1LM. Collectively, these analyses

provide useful insights into marker genes that are potentially involved in regulating the balance of the cell number in the SG.

*Increase in cell size*. Based on the enrichment analysis, we identified four cell types in the SG (ECRs, FPCs, CMs, and SPCs) that were enriched in genes related to the mTOR, InR, PI3K/Akt pathways (Fig. 6d; Supplementary Data 12), suggesting that genes in these pathways were involved in regulating cell size in the SG. Notably, the mTOR pathway positively regulates a large number of cellular processes, including growth, autophagy, mitochondrial biogenesis and lipid biosynthesis[46]; the InR pathway regulates carbohydrate and lipid metabolism, tissue growth, and longevity[47], and the PI3K/Akt promotes insulin-stimulated glucose metabolism and cell survival[48]. Among the enriched genes (Supplementary Data 11), *Akt1* was expressed in DRRs, ECRs, FPCs, CMs and SPCs, suggesting that it may serve as a link between the InR and TOR pathways and promote cell growth, as previously reported[47]. *CycG*, which has been proposed to positively regulate *Akt1* activity[49], was expressed at high levels in approximately the same cell types (DRRs, ECRs, FPCs, and SPCs) and might promote cell growth in the SG via *Akt1*. In addition, some other marker genes that may participate in regulating cell size were also identified in SG cells, but require further experimental verification.

*Intercellular communication and remodeling*. Based on the enrichment analysis, we found that some marker genes were enriched in "Gap junctions", "Tight junctions", and "Adherent junctions" (Fig. 6e; Supplementary Data 12), suggesting that these genes may be closely related to intercellular communication between SG cells. For example, *ASH1* functions in tight junctions between cells by binding the cells together and preventing molecules and ions from crossing the intercellular space[50], and was expressed in ECRs and FPCs. The *vinculin* gene detected in ECRs, FPCs, CMs, and SPCs, is a classical component of focal adhesions and adhesion junctions and functions in maintaining the structure and polarity of certain cells and limiting their movement and proliferation[51]. For the genes involved in the intracellular remodeling of SG cells, we mainly focused on those expressed in cell types that are specifically detected or abundant at 1LM, a period of vigorous cell metabolism and organ remodeling. As expected, numerous candidate genes were identified in the ECRs of the ASG, SPCs of the MSG, and FPCs of the PSG (Supplementary Data 12). Remarkably, many core members of the 20E signaling pathway were highly expressed at 1LM, including its receptor EcR and response factors Hr3, E74, Br-c, and Ftz-f1. The crucial roles of these factors in regulating intracellular processes such as cell autophagy, cell dissociation, and cell regeneration in other tissues of *B. mori* have been studied in depth; therefore, we were not surprised that the 20E signaling pathway is the key regulator of intracellular remodeling of SG cells. In addition, some other pathways related to cell autophagy were also identified, including the Hippo, MAPK, and Lysosome pathways (Fig. 6f; Supplementary Data 12). Although experiments are necessary to further verify their roles in remodeling SG cells, these results collectively provide important insights into the regulation of SG cell growth during early development.

**Spatiotemporal regulation of silk protein synthesis**. Of the three functional regions of the SG, only the MSG and PSG are known to have the ability to synthesize silk sericin and fibroin, respectively. However, the genes involved in the regulation of silk protein synthesis have never been analyzed at the single-cell level. To gain deeper insights into the spatiotemporal regulation of silk

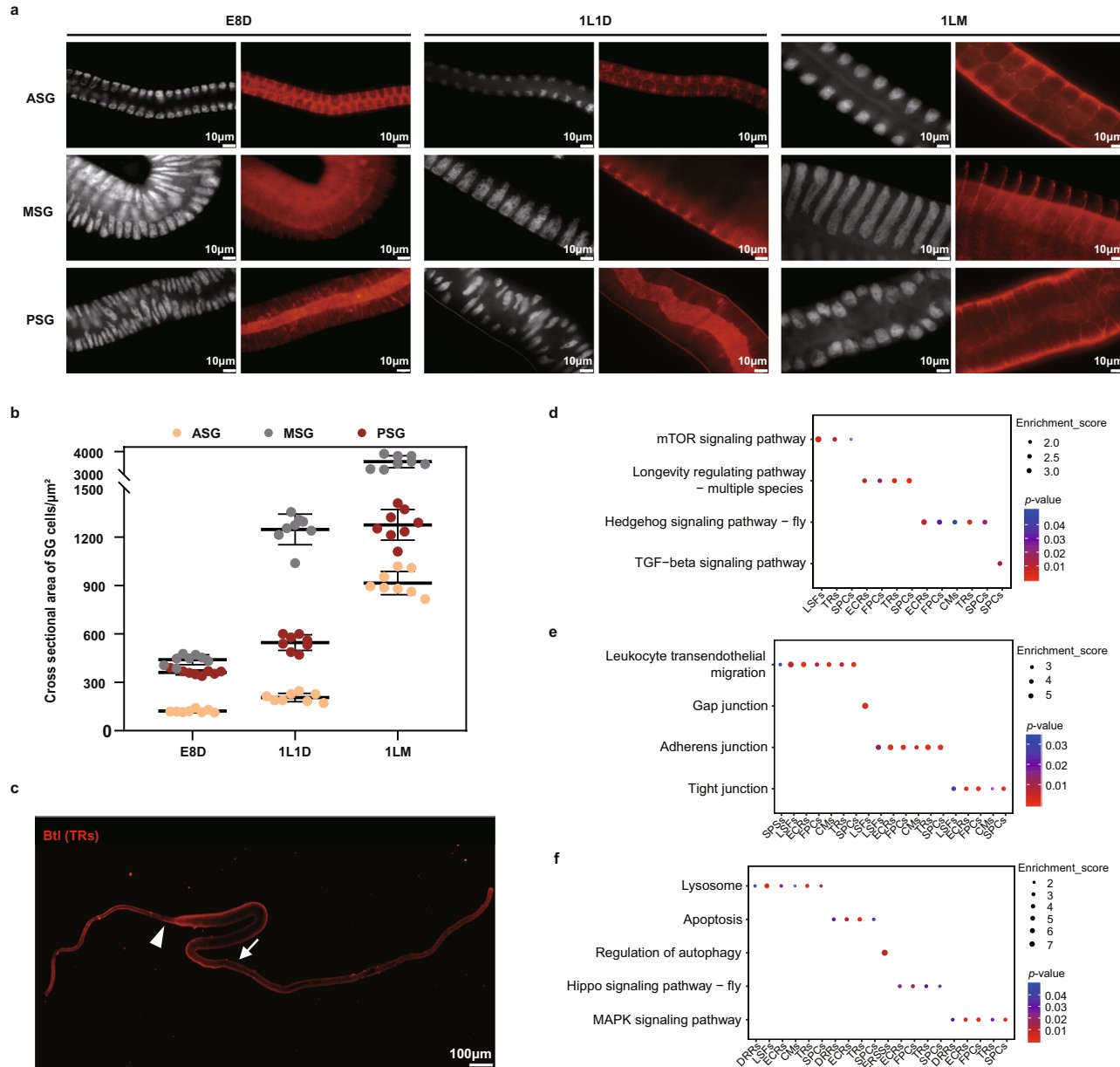

**Fig. 6 Effective strategies for inducing silk gland cell growth. a** Immunostaining of silk gland cells dissected from E8D, 1L1D, and 1LM individuals. Nuclei were stained with DAPI (white); the cytoskeleton was stained with phalloidin (red). The scale bar represents 10 μm. Representative images are chosen from photos of at least 10 intact silk glands. **b** The cross-sectional area of ASG/MSG/PSG cells was calculated from E8D to 1LM. Data are presented as mean ± SD ($n = 8$). Source data are provided as a Source Data file. **c** Immunofluorescence staining for Btl in 1L1D individuals. The spatial expression pattern is shown in red. The white arrowhead indicates the boundary between the ASG and MSG; the white arrow indicates the boundary between the MSG and PSG ($n \geq 10$ in three independent experiments). **d**–**f** Enriched GO/KEGG terms ($p < 0.05$) related to cell growth (**d**), communication (**e**) and renewal (**f**), $p$ value was calculated by the hypergeometric distribution, $p$-adj was obtained after $p$ value is corrected by Benjamin & Hochberg multiple test. Color gradients indicate the range of $p$ values. The $p$-values of each pathway are provided in Supplementary Data 12. ASG anterior silk gland, MSG middle silk gland, PSG posterior silk gland. E8D, 8 days post egg laying, 1L1D day 1 of the first instar, 1LM first larval molting.

protein synthesis, we analyzed the markers in those clusters belonging to the MSG and PSG.

Within the cell types of MSG, the sericin protein-coding genes *Ser1* and *Ser2* were expressed at high levels in SPSs and SPCs, while *Ser3* was not identified as a marker gene in any of the 10 cell types. We found that *Ser3* was expressed at very low levels in SPSs and SPCs (Fig. 7a, Supplementary Data 13), which is not surprising because *Ser3* has been proven to be expressed abundantly only in the MSG of last instar larvae[52]. Immuno-fluorescence staining further confirmed the distribution of Ser1, Ser2, and Ser3 (Fig. 7b). *Ser4*, a newly identified gene encoding

sericin protein 4[53], was expressed at high levels in SPSs and SPCs. These results indicate that the synthesis of the sericin proteins is differentially regulated in MSG cells. In particular, SPSs seem to be the major cell type synthesizing sericin, as this cluster expresses all three genes (*Ser1*, *Ser2*, and *Ser4*) at E8D, 1L1D, and 1LM. The GO analysis of SPSs showed that marker genes of this cluster were enriched in "Formation of the cytoplasmic translation initiation complex" and were mainly different splice isomers of eukaryotic translation initiation factor 3 (eIF3) that play a central role in regulating protein synthesis[54] (Fig. 7c; Supplementary Data 14). *Fkbp46* is another interesting marker of

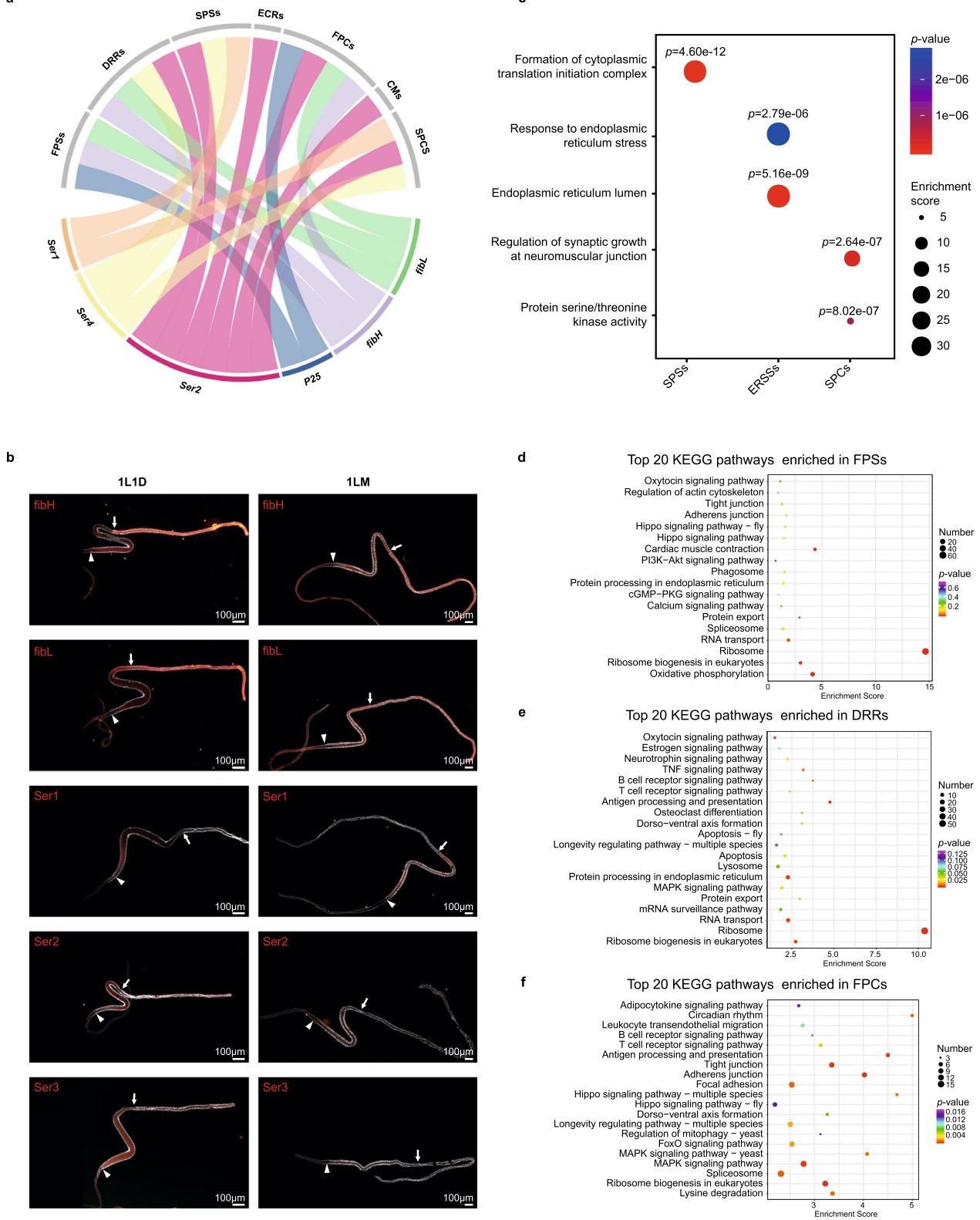

SPSs that may be involved in regulating sericin synthesis, as it not only binds DNA but also binds proteins through its acidic domain[55]. SPCs are considered to play important roles in the MSG during molting, and many members or downstream targets of the 20E and Hippo pathways were expressed at high levels in SPCs, including *E75*, *Br-c*, *Hr3*, *Hr39*, *Hr4*, and *Ftz-f1* in the 20E signaling pathway and *yki*, *Diap1*, and *E2F1* in the Hippo

signaling pathway (Supplementary Data 1). Interestingly, *sage* was expressed at high levels in SPSs and ERSSs. Sage regulates the regional specificity of silk protein gene expression by competing with SGF1 for FKH-1 and thereby negatively regulating *Dfd*[56], suggesting that *sage* is involved in sericin protein synthesis. Surprisingly, *Ser2* was also detected in ASG (i.e., ECRs and CMs) and PSG (i.e., DRRs and FPCs) cells, and *Ser4* was observed in

**Fig. 7 Spatiotemporal regulation of silk protein synthesis. a** Chord diagram of the distribution of major silk protein-coding genes in each cell type.
**b** Immunofluorescence staining for silk protein-coding genes in 1L1D and 1LM. The red part of the silk gland shows the location of the protein distribution.
The white arrows demarcate the three regions of the silk gland. Nuclei were stained with DAPI ($n \geq 10$ in three independent experiments). 1L1D, day 1 of the
first instar; 1LM, first larval molting. **c** Representative GO terms enriched in sericin protein-synthesizing cells (SPSs), endoplasmic reticulum stress signaling
cells (ERSSs), and sericin protein catabolism cells (SPCs) that regulate sericin protein synthesis. *P* values are shown in the charts are determined by
multiple hypothesis testing. Source data are provided in Supplementary Data 14. **d–f** Top 20 KEGG pathways enriched in fibroin protein-synthesizing cells
(FPSs), death and remodeling regulation cells (DRRs), and fibroin protein catabolism cells (FPCs) that regulate fibroin protein synthesis ($p < 0.05$). *P*-value
was calculated by the hypergeometric distribution. Color gradients indicate the range of *p*-values. The *p*-values of each pathway are provided in
Supplementary Data 15.

PSG cells (i.e., DRRs). We speculate that the expression of these
two sericin genes is not strictly regulated during early develop-
mental stages or that they play unknown but crucial roles in the
ASG and PSG of early developmental stages.

Within the cell types of PSG, the fibroin protein-coding genes
*fibH*, *fibL*, and *P25* were all expressed in FPSs, DRRs, and FPCs
(Fig. 7a). The expression levels of these three genes were
significantly different in the three developmental stages, with
the highest expression levels in 1L1D and the lowest in E8D.
Among them, *fibH* exhibited the highest expression level, and *P25*
exhibited the lowest (Fig. 3a). Immunofluorescence and fluores-
cence expression analyses based on transgenic GAL4/UAS lines
further confirmed the distributions of *fibH*, *fibL*, and *P25*
(Figs. 3b; 7b; Supplementary Fig. 7). As 1L1D larvae spin silk,
we then focused on FPSs (the unique cell type enriched in 1L1D)
to identify the regulatory genes involved in fibroin protein
synthesis and identified a large number of ribosomal genes
enriched in the "Ribosome" term (i.e., *RpL3–19* and *RpS3–21*)
(Fig. 7d, Supplementary Data 15), suggesting that these genes
may contribute to the efficient synthesis of fibroin protein
because protein synthesis requires the participation of a number
of ribosomal genes. Notably, several TFs were also identified in
FPSs, including Ybp, ALY, Rack1, and LOC692655, which may
participate in the transcriptional regulation of fibroin protein
synthesis. An interesting finding was that fibroin genes were
expressed in the PSG cells of 1LM individuals (DRRs and FPCs),
and many TFs that have previously been shown to regulate
fibroin genes were detected in DRRs and FPCs, including SGF1,
sage, and core members of the 20E signaling pathway (EcR, Br-c,
Hr3, E74, E75, and Ftz-f1). In addition, the core members of the
Hippo and MAPK signaling pathways, including yki, sd, Myc,
Diap1, 14-3-3, and Ras, were also identified in DRRs and FPCs
(Fig. 7e, f; Supplementary Data 15). Therefore, many TFs were
expressed in the PSG cells of 1LM rather than 1L1D, reflecting the
complexity of the stage-specific regulation of fibroin protein
synthesis. Moreover, it seems more important to accurately
inhibit fibroin protein synthesis during early developmental
stages, which is an interesting biological phenomenon worthy of
further study.

## Discussion

In this study, we report a comprehensive single-cell tran-
scriptomic atlas of 14,972 high-quality cells filtered from three SG
samples of *B. mori* during early development. This atlas is com-
posed of 10 distinct cell types that correspond to three regions of
the *B. mori* SG. Our work contributes valuable information about
(1) the heterogeneity of SG cells at early developmental stages, (2)
the possible biological functions of each cell type in the three
regions of the SG, (3) the developmental trajectories of SG cells,
(4) the gene expression status in SG cell types, and (5) the spa-
tiotemporal regulation of SG development and silk protein
synthesis during the embryonic and early larval stages. These
findings provide insights into the SG at single-cell resolution.

The SG of *B. mori* has obvious divisions in structure and
function and is generally divided into three anatomically and
physiologically distinct regions, the ASG, MSG, and PSG[12]. Only
the MSG and PSG are known to synthesize sericin and fibroin
proteins (two major components of silk fiber), respectively, but
which cells synthesize these proteins remain unknown. In this
study, we determined the distribution of the major cell types in
each region of the SG. Specifically, the ASG consisted of LSFs,
ECRs, CMs, and TRs, the MSG was composed of SPSs, ERSSs,
and SPCs, and the PSG consisted of FPSs, DRRs, and FPCs. Of
the genes encoding fibroin proteins, *fibH* and *fibL* were expressed
at high levels in FPSs, DRRs, and FPCs, and *P25* was highly
expressed only in FPSs and FPCs, consistent with our experi-
mental results. Among the major genes encoding sericin proteins,
*Ser1* and *Ser2* were identified as markers of SPSs and SPCs, but
not the E8D-specific ERSSs. Experimental evidence supported
this observation and implied that ERSSs mainly perform func-
tions other than sericin protein synthesis in the embryonic MSG.
*Ser3* was not identified as a marker gene in any of the 10 cell
types. Experimental analysis showed that *Ser3* was expressed at a
very low level in the MSG of 1L1D and hardly detected in E8D
and 1LM, suggesting that *Ser3* activity was severely inhibited in
the MSG of silkworms early in development. Surprisingly, *Ser2*
and the recently described sericin protein-coding gene *Ser4*[53],
were also expressed at high levels in PSG cells (i.e., DRRs).
Considering previous evidence that the Ser2 and Ser4 proteins are
not present in cocoon silk[23,53], the observed high expression of
*Ser2* and *Ser4* in both MSG and PSG cells implies special roles for
these proteins in the SG of silkworms during early development;
for example, they are likely to contribute to the mechanical
properties of silks spun by early silkworm larvae, which deserves
further study. Together, these findings clearly reveal the hetero-
geneity of SG cells and reflect that different cell types in the SG
play different roles in silk protein synthesis.

Another interesting issue is the gene expression profiles in
distinct cell types, which is an important foundation for under-
standing their biological functions in the SG. In view of this, we
performed deep analysis, including pseudotime and gene-switch
analysis. Many genes expressed in the cell types that were enri-
ched in the SG of the first instar larvae were inactivated in E8D
cells. Functional terms involved in developmental regulation were
not enriched in the cell types characteristic of the SG in early
development, including ERSSs (E8D), SPSs (E8D-1LM), and FPSs
(E8D and 1L1D). These observations reflect the low activity of
genes that are expressed at high levels during larval stages and
suggest a functional transition of SG cells from the embryonic to
larval stage, a period in which significant morphological changes
occur in *B. mori*. Notably, most of the activated genes existed in
the SG at 1LM. In insects undergoing metamorphosis, 20E sig-
naling is a vital regulator of larval molting and metamorphosis[57].
We found that many core members of the 20E signaling pathway,
including *EcR*, *Br-c*, *Hr3*, *Hr4*, *E74*, *Hr38*, *Hr39*, and *Ftz-f1*, were
expressed at high levels mainly in SG cells at 1LM (i.e., ECRs,
CMs, and SPCs). Among these genes, upregulated expression of

*Ftz-f1* and *Hr39* is essential for tissue removal by initiating cell apoptosis during metamorphosis, while *EcR*, *E74, E75* and *Br-c* induce autophagy to remodel larval tissues[58]. In addition, "Lysosomes" and "Autophagy" were also enriched during ecdysis. These results suggested that extensive autophagy was initiated by 20E signaling in 1LM SG cells, and thus promoted the recycling of intracellular substances and cell renewal. 20E signaling has been also shown to negatively regulate the expression of silk protein genes[59,60]. We found that the expression of silk protein genes in the SG was significantly higher at 1L1D than at 1LM, strongly suggesting an inhibitory role for 20E signaling in silk protein synthesis in SG cells during larval molting. In addition, we also identified numerous key TFs (i.e., SGF3, sage, Awh, and dimm) and signaling pathways involved in the regulation of the SG at 1LM. For example, core members of the Hippo pathway (i.e., sav, wts, yki, and sd), a key conserved signaling pathway that controls organ size during development by restricting cell growth and proliferation and promoting apoptosis[38], were found to be highly expressed in FPSs, DRRs, and CMs at 1LM. These results imply that in SG cells, 20E signaling may cooperate with Hippo to jointly control SG development and silk protein synthesis during larval molting. Collectively, our analysis of the expression profiles of marker genes further revealed the heterogeneity of the SG cells. In-depth clarification of the identified marker genes will further promote the understanding of the regulatory mechanisms of SG development and silk protein synthesis.

In summary, we have built a single-cell atlas of the *B. mori* SG, which is composed of 10 distinct cell types distributed in the ASG, MSG, and PSG, three physiologically distinct regions of the SG, and revealed their developmental trajectories, gene expression status, and the representative marker genes involved in SG development and silk protein synthesis (Fig. 8). This study will help drive discovery forward by providing a deeper understanding of the cell composition and heterogeneity of the *B. mori* SG. The transcriptional atlas of each cluster is an informative dataset that can be used to explore the biological functions of relevant cell types. Furthermore, these efforts will provide a valuable resource for future studies that use the *B. mori* SG as an experimental model and thus accelerate studies on the development of silk-producing organs, the mechanism of silk protein synthesis, and even the genetic modification of silk using cell-type-specific marker genes.

## Methods

**Single-cell suspension preparation and RNA sequencing**. *B. mori* (*Nistari* strain, which was obtained from Silkworm Gene Bank of Southwest University, Chongqing, China) embryos developed to E8D, 1L1D larvae, and 1LM larvae were maintained under standard laboratory conditions and collected to obtain SG samples. SG samples from each developmental stage were composed of a mixture of ~2000 intact SGs to ensure sufficient materials for subsequent cell dissociation. Tissues were separately dissociated and then resuspended in 1 mL PBS (Sangon Biotech, China) containing 0.04% BSA (Beyotime, China). After the cell viability test, single-cell separation, complementary DNA amplification and library construction were performed according to the manufacturer's protocol for Chromium Single-cell 3′ Kits (10× Genomics). The libraries were then sequenced on the Illumina sequencing platform (NovaSeq 6000), and 150 bp paired-end reads were generated.

**Data preprocessing and analysis**. The Cell Ranger software pipeline (version 3.1.0) provided by 10× Genomics was applied to demultiplex the cellular barcodes, map reads to the reference genome (https://ftp.ncbi.nlm.nih.gov/genomes/all/GCF/ 000/151/625/GCF_000151625.1_ASM15162v1/GCF_000151625.1_ASM15162v1_ genomic.fna.gz) and transcriptome using the STAR aligner, and downsample reads as required to generate normalized aggregate data across samples, generating a matrix of gene counts versus cells. Then, the unique molecular identifier (UMI) count matrix was fed into the R package Seurat[16] (version 3.1.1) for downstream processing. To remove low-quality and likely multiplet cells, which is a major concern in microdroplet-based experiments, we assumed that the ratio of UMI counts versus gene counts per cell would follow a Gaussian distribution and estimated the mean and standard deviation. Any cell with a ratio that deviated from the

mean value by at least two s.d. was discarded. Further, any cells with >30% of counts from mitochondrial genes were removed. After these quality-control steps, single cells were retained for downstream analyses. Library size normalization was then performed using the *NormalizeData* function in Seurat to obtain the normalized count. Specifically, the global-scaling normalization method "LogNormalize" was used, where gene expression was normalized by the total expression of each cell and multiplied by a scaling factor (10,000 by default), with the results log transformed.

Next, the most variable genes were selected using the *FindVariableGenes* function (mean.function = FastExpMean, dispersion.function = FastLogVMR)[61]. Batch effects were removed by applying the mutual nearest neighbors (MNN) approach [Haghverdi[62]] using the batchelor package in R. Cells were then clustered according to their gene expression profile by using the *FindClusters* function in Seurat. Cells were visualized by using the 2-dimensional uniform manifold approximation and projection (UMAP) algorithm, which further reduces dimensionality, with the Seurat function *RunUMAP*. The Seurat Findallmarker function (test.use = bimod, logfc.threshold = 0, min.pct = 0.25) was used to identify preferentially expressed genes between clusters. Especially, Padj was calculated based on Bonferroni correction. Differentially expressed genes (DEGs) for clusters of interest were identified using the *FindMarkers* function (test.use = MAST, min.pct = 0.1). A $p$ value < 0.05 and |$\log_2$ fold change |> 0.58 were set as the thresholds for statistically significant differences. Detailed information corresponding to gene IDs of all marker genes mentioned in the text are summarized in Supplementary Data 16. GO enrichment and KEGG pathway enrichment analysis of DEGs were respectively performed using R based on the hypergeometric distribution.

**Pseudotime analysis**. Single-cell developmental pseudotime was determined using the R package Monocle2 (version 2.9.0)[25]. The raw count matrix was first converted from a Seurat object into a CellDataSet object with the *importCDS* function in Monocle, and the *differentialGeneTest* function in Monocle2 was then used to select genes (qval < 0.01) that were likely to be informative for ordering cells along the pseudotime trajectory. Dimensional reduction clustering analysis was performed with the *reduceDimension* function, followed by trajectory inference using the *orderCells* function with default parameters. Gene expression was plotted by the *plot_genes_in_pseudotime* function to track changes over pseudotime.

**RNA velocity analysis**. The spliced and unspliced reads were recounted using the Python script velocyto.py[63] (https://github.com/velocyto-team/velocyto.py) to conduct the RNA velocity analysis. The RNA velocity values for each gene in each cell and the RNA velocity vector with reduced dimensionality were calculated using the R package velocyto.R (version 0.6). Velocity fields were projected onto the UMAP plot generated in the previous step.

**Gene-switch analysis**. Gene-switch analysis was performed to discover the order of gene expression and other functional events during cell state transitions at single-cell resolution. First, gene expression data were binarized into 1 (on) or 0 (off) states with the *binarize_exp* function in the Gene-switch package[64] (version 0.1.0) (fix_cutoff = TRUE, binarize_cutoff = 0.05). Specifically, a mixture model of two Gaussian distributions was fitted for each gene to calculate gene-specific thresholds for binarization. Prior to fitting, Gaussian noise with a mean of zero and standard deviation of 0.1 was added to the gene expression, which ensured numerical stability in the fitting of the gene expression. Genes that did not have a distinct bimodal "on-off" distribution were then removed. Logistic regression was then applied to model the binary states (on or off) of gene expression with the *find_switch_logistic_fastglm* function. Random downsampling of zero expression (downsample = TRUE) was used to rescue genes with high zero inflation for the prediction of switching time. Finally, the top 50 best fitting (high McFadden's Pseudo R^2) genes were plotted along the pseudotime line.

**qRT–PCR analysis**. ASG, MSG, and PSG samples were isolated from the E8D embryos and 1L1D and 1LM larvae. Total RNA was extracted from each of the SG samples using the E.Z.N.A. MicroElute Total RNA Kit (Omega Biotek, USA). cDNA templates were obtained through reverse transcription of 0.4 μg of total RNA using PrimeScript™ RT Reagent Kit (Takara, Japan). The mRNA levels of marker genes were quantified by qRT–PCR using SYBR® Premix Ex Taq™ (Takara, Japan) on QTOWER 2.0 (ANALYTIKJENA, Germany). A 20 μL mixture of 2 μL of cDNA templates, 0.8 μL of upstream and downstream primers, 10 μL of SYBR® Premix Ex Taq™, 0.4 μL of ROX Reference Dye, and 6 μL of ddH$_2$O was reacted at 95 °C for 30 s for pre-denaturation; followed by 40 cycles of 95 °C for 3 s and 60 °C for 30 s; then 95 °C for 15 s, 60 °C for 1 min, and 95 °C for 15 s. *B. mori* eukaryotic translation initiation factor 4 A (*BmeIF4A*)[65] was used as an internal control. The primers used for qRT–PCR are listed in Supplementary Table 1. All experiments were carried out with three biological replicates, and more than 200 silkworm individuals were used in the experiment. All analyses were completed using GraphPad Prism 8.0.

**Immunofluorescence analysis**. The intact SGs were dissected and washed with 1 × PBS (Sangon Biotech, China), placed on a slide for fixation with 4% paraformaldehyde

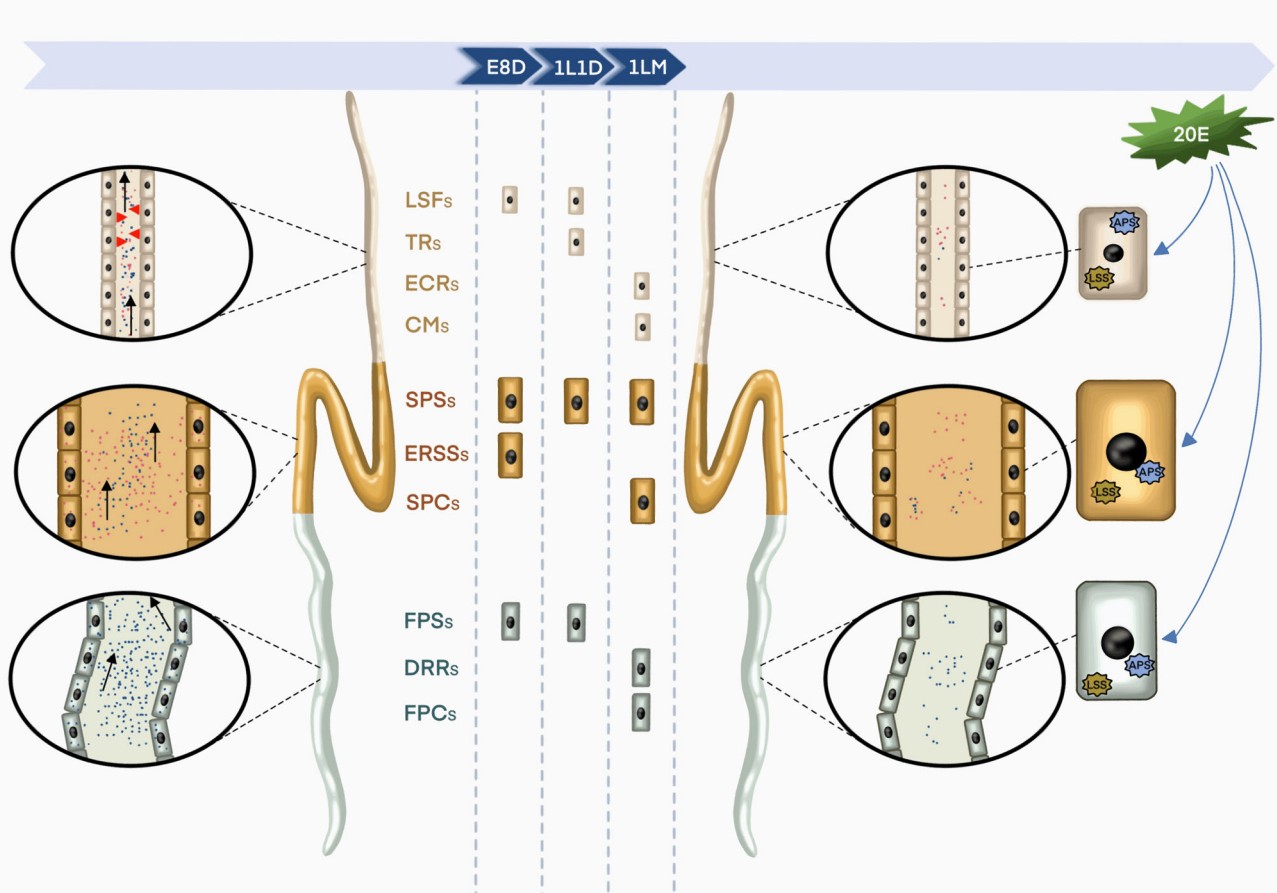

**Fig. 8 Depicting the composition of major cell types in silkworm silk glands.** Cell types in the anterior silk gland, middle silk gland, and posterior silk gland are distinguished in light brown, gold, and white, respectively. Liquid silk fibrosis cells (LSFs), sericin protein-synthesizing cells (SPSs), endoplasmic reticulum stress signaling cells (ERSSs), and fibroin protein-coding synthesizing cells (FPSs) appeared at E8D; LSFs, SPSs, FPSs, and traction forces cells (TRs) appeared at 1L1D; SPSs, FPSs, epithelial cell remodeling cells (ECRs), chitin metabolism cells (CMs), sericin protein catabolism cells (SPCs), death and remodeling regulation cells (DRRs), and fibroin protein catabolism cells (FPCs) appeared at 1LM. Interestingly, ERSSs were E8D-specific, TRs were 1L1D-specific, and ECRs, CMs, SPCs, DRRs, and FPCs were 1LM-specific. The black arrow represents the direction of silk protein transport and the red arrowhead represents shear force. Magnified images of the corresponding silk glands are shown in black panes (both sides). An enlarged view of single cell at 1LM modulated by 20E signaling is shown on the right. Blue dots: silk fibroin; red dots: sericin; APS: autophagosome; LSS: lysosome. E8D 8 days post-egg laying, 1L1D day 1 of the first instar, 1LM first larval molting.

(Sangon Biotech, China), and permeabilized with Immunostaining Permeabilization Solution with Triton X-100 (Beyotime, China). The slides were then blocked with 5% fetal bovine serum (Beyotime, China) for 30 min, incubated with primary antibodies, all of which were commercially prepared by Zoonbio Biotechnology Co., Ltd, China, including fibH (#ZD17N03M5383), fibL (#ZD17N03M5385), P25 (#ZD17N03M5386), Ser1 (#ZD2018V002945), Ser2 (#ZD2018B002943), Ser3 (#ZD2018P002944), LOC101746180 (#ZD2021O013151), LOC101740197 (#ZD2021P013154), LOC101745308 (#ZD2021D013149), Btl (#ZD2021C013155) at a dilution ratio of 1:150 for 2 h, and then incubated with secondary antibody of anti-rat IgG (H + L) Alexa Fluor® 555 Conjugate (#4417, CST, USA) at a dilution ratio of 1:150 for 1 h. Finally, the nuclei were stained with DAPI solution (Beyotime, China), and the slides were sealed with Antifade Mounting Medium (Beyotime, China) for observation. The fluorescence images were observed and analyzed using an inverted fluorescence microscope (Olympus, Japan). The antibody information is listed in Supplementary Table 2.

**Transgenic expression analysis**. Transgenic silkworms harboring UAS-linked EGFP (named UEGFP) and the GAL4 transgenic silkworm strains fibH-GAL4 (HG4), fibL-GAL4 (LG4), and P25-GAL4 (PG4), in which *GAL4* gene expression was controlled by the promoters of the silk fibroin protein-coding genes *fibH*, *fibL*, and *P25*, respectively, were generated previously by our group[66] and maintained in the State Key Laboratory of Silkworm Genome Biology, Southwest University. The hatched silkworm larvae were reared normally on mulberry leaves at 25–26 °C. Each of the GAL4 silkworms was separately crossed with UEGFP to produce and screen the corresponding offspring, termed HG4/UEGFP, LG4/UEGFP, and PG4/UEGFP. Then, GAL4/UAS individuals developed to E8D, 1L1D, and 1LM were

randomly selected for dissection of the SG and detection of the EGFP expression patterns using a fluorescence stereomicroscope (Olympus, Japan).

**Statistics and reproducibility**. GraphPad Prism version 8.0 was used for statistical analysis in Figs. 3a and 6b. Data are shown as mean values ± SD, three independent experiments were conducted and shown as data points in graphs. The microscopic images in Figs. 1b, 3b, 6a, c, 7b, and in Supplementary Figs. 2 and 7 are representative of more than five individuals. No statistical methods were used to pre-determine sample sizes. All data sets were included in this manuscript. Randomization and blinding were used.

### Data availability
The single-cell RNA sequencing data generated in this study have been deposited in the NCBI GEO database under accession code "GSE193279" and in SRA datasets under BioProject "PRJNA795657". Raw sequences from each sample are available under accession number PRJNA662018, with the following login numbers: "SAMN24744553", "SAMN24744554" and "SAMN24744555". All other relevant data supporting the key findings of this study are available within the article and its Supplementary Information files or from the corresponding author upon reasonable request. Source data are provided with this paper.

### Code availability
R Scripts for QC, clustering in Seurat (version 3.1.1) and Monocle2 (version 2.9.0) used for the pseudotime analysis are all provided in Supplementary Software 1 and codes are available in the "Github repository [https://github.com/superly917/silk_gland_atlas]"[67].

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

## Acknowledgements

This work was supported by grants from the National Key Research and Development Program of China (project number 2021YFE0111100) to H.F.X., the National Natural Science Foundation of China (project number 31872291) to H.F.X., the Academician Foundation Program of Chongqing (project number cstc2020yszx-jcyjX0003) to H.F.X., the Talents Project of Chongqing (project number cstc2021ycjh-bgzxm0047) to H.F.X., and the Science and Technology Research Program of Chongqing Municipal Education Commission (project number KJQN202100211) to Y.M.

## Author contributions

Y.M., W.H.Z., Q.L., Y.O., R.P.L., J.W.M., Y.Y.M., Q.J.L., Y.Q.C. and Y.T.R. performed the experiments. Y.O., R.P.L., Y.Y.T., J.H. and H.M.W. prepared the silkworm eggs and larvae used for SG sampling. Y.B.B., X.T., Y.M., Q.L. and Y.O. performed the bioinformatics analysis. Y.M., W.H.Z., Y.B.B., Q.L., Y.O. and H.F.X. wrote the manuscript. Y.M., Z.H.X. and H.F.X. conceived the study.

## Competing interests

The authors declare no competing interests.

## Additional information

**Peer review information** *Nature Communications* thanks Kakeru Yokoi, and the other, anonymous, reviewer(s) for their contribution to the peer review this work. Peer reviewer reports are available.

