## [Peer Review File · Nature Communications]

REVIEWER COMMENTS

Reviewer #1 (Remarks to the Author):

In this study, the authors performed single cell RNAseqs using silk grand (SG) cells of E8D, 1L1D and 1LM. They revealed SG consisted of 10 types of cells according to the transcriptome profiles, and they determine whether each cell type mapped to region of the SG, which are SG, MSG and PSG by utilizing classic maker gene and cluster-specific genes. To reveal the characteristics of each cell types, they performed spatiotemporal analysis using top 10 maker genes and showed SG is high heterogeneity, at early stages. They performed pseudotemporal analysis and gene-switch analysis using the single cell RNAseq data in the three region of SGs, and several new insights into SG cells were revealed. Consequently, they identified candidate genes involved in SG cell growth using the atlas of transcriptomes of SG cell and gene annotation data. Finally they revealed more deeper insights of silk protein synthesis using these atlas. Through the series of analysis in this manuscript, the authors provide new and detailed insights into SG cell of silkworms.

In my opinion, the authors reveal new and attractive knowledge of SGs encourages some researchers to do further analysis to get more detailed SGs. As they described in introductions, silkworms is important insects for both textile industry and basic biology as model Lepidopteran species. Considering them, this manuscript is worth to publication of Nature communication. However, there are several points which they should address especially in methods.

1. They did not show accession ID of each raw sequence data of single RNAseqs. In line 610, they showed only GEO code. For reproducibility, they must ensure the accessibility of each raw sequence by showing accession ID through depositing the raw data in public database.
2. Several versions of data analyzing tools were not shown. For examples, the version of Monocles2 in line 546 was not shown. Authors must show versions of all packages they used.
3. About Gene-switch analysis (from line 558), they did not show any references or URLs. If several URL or references do not exist, they must upload all of codes they perform in public repository (e.g. Github).
4. Detailed conditions of qRT-PCR reactions were not shown . (In line 570-577).
5. The resolution of fig.5 is very low. Replace them to one with high resolutions.
6. They used several gene IDs in this manuscript. However, detailed information of these gene set data they used was not shown. Which data? Version and accessing points must be shown in method section.
7. For more clearly understanding the author's works, I strongly recommend to add Scheme diagram of data analysis in the end of introduction. Also, summary diagram of the manuscript must be added in discussion part. The readers must easily understand your works.

Minor point

1. Fig 6b is poor recognizable because of complicated colors. Change it to dash style or other expression style.
2. Fig 5, alphabetical orders in figure and main text are incorrect.

I hope that my comments help improving your manuscript.

Reviewer #2 (Remarks to the Author):

In this manuscript, Ma and colleagues attempted to identify the cellular atlas of silk glands in *Bombyx mori* and to elucidate the molecular mechanisms involved in silk protein production at the single-cell level by single-cell sequencing. Overall, this manuscript is novel and interesting in terms of experimental purpose and experimental technique. However, there are still some problems in this manuscript in terms of bioinformatics analysis and description of results. Authors need to make substantial revisions to be considered for publication.

Major Points and Questions:

1. The samples selected by the authors is not representative. As described by the author, 'the majority of protein synthesis begins on day 3 of the fifth instar'. The author claims that there is no way to take this sample due to technical problems. In fact, the authors can achieve this time point analysis using single-nucleus techniques. The lack of samples at this time point could have important implications for the description of the results of this study. Although I don't know much about the silkworm and the silk spinning mechanism, it is obvious that it is quite difficult to reveal the molecular mechanism of silk protein production from the results obtained by the author at the time point of the sample.
2. The author simply divided silkworm silk gland cells into 10 clusters based on bioinformatics. In fact, as long as we adjust the parameters, the number of cell clusters will change, and the entire analysis results will change. Why did the authors not identify silk gland cells? I think the authors should try to identify SG cell types. Of course, it is difficult for non-model organisms, but the work still needs to be done. Clusters would be much easier to follow using cell type names.
3. The results of the trajectory of SG development accounts for a large proportion of the full manuscript. However, the authors do not specify which cells are the real starting point. In fact, if the authors did not identify the SG clusters, it would be impossible to know which cell types belonged to the early differentiated cells. If the exact starting point of differentiation is not specified in pseudotime analysis, the subsequent analysis results are less credible.
4. In this manuscript, the authors use Immunostaining to detect marker genes. The antibodies used by the authors are polyclonal antibodies, from Zoonbio Biotechnology (Supplementary Table 2). The effect of this polyclonal antibody seems to work well. Is this result reliable and will there be false positives? Is it a commercial antibody or a self-made one? The author needs to mark clearly in the text. And the

country and region to which the company belongs also needs to be indicated. My concern is that generally everyone uses Fluorescent in situ hybridization (FISH) or in situ hybridization (ISH) to validate marker genes. Why don't the authors use this method?

Minor Points and Questions:

1. The author emphasizes the use of 2000 intact SGs for each sample. How do you ensure that such a large number of samples are collected in a short period of time while still maintaining their validity.

2. Which reference genome did the authors use? The authors did not indicate in the Methods section (line 521). Two genome IDs appear in the text. It's puzzling. Line 126 ,BGIBMGA011721 and others LOC. LOC seems to be from NCBI (Bmori_2016v1.0 ?), why didn't the authors use the latest published genome of the silkworm as the reference genome (doi: 10.1016/j.ibmb.2019.02.002 OR doi: 10.1093/nar/gkz919)?

3. In figure 1d, the UMAP is not as clear as in other articles. The UMAP results in this manuscript look a little out of the ordinary. Although the author removes likely multiplet cells by some methods in the text. Has the author considered further removal of polycells through tools such as DoubleFinder?

4. Line 186 and 187, Authors should only select insect libraries for KEGG analysis, not vertebrate data. These results are clearly contrary to common sense and do not deserve further elaboration.

5. Line 191-192, I think the 'above results' are not accurate to demonstrate the high heterogeneity of SGs. It's normal for a single organ to be made up of many kinds of cells. Heterogeneity requires objects of comparison.

6. Line 511, Why did the authors not choose the silkworm strain DAzao, whose genome has already been published.

7. Many conclusions in the text have overstate. Authors need to pay attention to accurate description of results.

8. The resolution of some pictures needs to be improved, as shown in Figure 5b, d, f.

Reviewer #3 (Remarks to the Author):

In this study, the authors conducted single-cell transcriptomic analysis on the silkworm silk gland of three early developmental stages. Generally, this study provided new data and added our understanding of silk gland development at early developmental stages, which is of potential value. However, I had expected much more than what the manuscript told at this step.

First of all, the whole manuscript is too much descriptive, with so many trivial findings but without compelling highlights or emphasis. I can't catch the novel point(s), from the manuscript. For example, As

to the result section “ Dynamic changes in gene expression in SG cells”, I’m really drown in the batch of turn-on and turn-off genes. Similar situation was also in “Regulation of SG growth and development in early stages”. What can the author infer from the massive description?

Furthermore, if “ mitotic division of SG cells ceases after stage 25 (6 days post-egg laying)¹², which means that the SG has completed morphogenesis at this stage”, then why new cell types appear or disappear at different developmental stages? I think maybe cell differentiation occurs during silk gland development. If this is the case, collection of more samples with different developmental stages, particularly the last larval stage when the mature silk gland may have distinctive characters compared with early-stage one. Regrettably there seems no good approach for SC-RNA seq on silk gland at this stage. This issue becomes difficult and hard to be addressed.

Specifically, Results of clustering and annotation of the cell types are not well evidentiary supported. In addition, the demonstration and explanation of the results are confusing.

1. Fig 1g and 1h, of course one can get specific or highly expressed genes in each cluster. The problem is how can we infer from these so-called marker genes? What is their biological significance ?
2. Line126-134. What’s the relationship of these genes with those shown in Fig 1g and 1h? Since they are all marker genes, the authors should not explain them separately. Specifically, the biological significance of the genes labeled by geneID and known markers in different regions of the silk gland should be demonstrated. Why these genes could annotate different region of the silk gland? By the way, the genes are all shown with gene ID but not gene name, hampering our understanding of their biological significance. Results of Immunofluorescence verification did not clearly supported the annotation. The authors need to explain the results.
3. I’m confused in the enrichment analysis. Only top 10 marker genes, how to generated enrichment analysis?
4. Cell clusters specific to certain region of the silk gland, could be better used for sub-cluster analyses.

Some statements are not precise, even wrong.

Line 483-485: “...suggest a functional transition of SG cells from the embryonic to larval stage, the first metamorphosis”. I think this is a wrong statement. I don’t think development from embryonic to larval stage is a process of metamorphosis. Particularly, at E8D, the embryo is factually well developed and similar to newly hatched silkworm larva.

As to results on hormone regulation on silk protein synthesis. As I know, there are many recent advances in this issue. The authors need to discuss the results deeply, consulting recent references rather than merely objective implication. For instance, Line 492-493:”These results strongly suggest that 20E signaling plays a crucial role in inhibiting silk protein synthesis in SG cells during larval molting”.

Other:

Description of sampling and sequencing quality is not clear. As I understand, totally three samples was subjected to single-cell RNA-seq, respectively? If this is the case, then how many cells sequenced in each sample? In addition, the reference genome information was not provided. Which version was used for reads mapping? How about the mapping rate?

Fig7 The first letter of the text should be capitalized.

Response to the Reviewers

Reviewer #1:

In this study, the authors performed single cell RNAseqs using silk gland (SG) cells of E8D, 1L1D and 1LM. They revealed SG consisted of 10 types of cells according to the transcriptome profiles, and they determine whether each cell type mapped to region of the SG, which are SG, MSG and PSG by utilizing classic marker gene and cluster-specific genes. To reveal the characteristics of each cell types, they performed spatiotemporal analysis using top 10 marker genes and showed SG is high heterogeneity, at early stages. They performed pseudotemporal analysis and gene switch analysis using the single cell RNAseq data in the three region of SGs, and several new insights into SG cells were revealed. Consequently, they identified candidate genes involved in SG cell growth using the atlas of transcriptomes of SG cell and gene annotation data. Finally they revealed more deeper insights of silk protein synthesis using these atlas. Through the series of analysis in this manuscript, the authors provide new and detailed insights into SG cell of silkworms.

In my opinion, the authors reveal new and attractive knowledge of SGs encourages some researchers to do further analysis to get more detailed SGs. As they described in introductions, silkworms is important insects for both textile industry and basic biology as model Lepidopteran species. Considering them, this manuscript is worth to publication of Nature communication. However, there are several points which they should address especially in methods.

Authors' response: Thank you very much for your kind and valuable comments, which provide great encouragement for our future work. Based on your comments and suggestions, we have carefully revised the manuscript, and our point-by-point responses are presented below.

1. They did not show accession ID of each raw sequence data of single RNAseqs. In line 610, they showed only GEO code. For reproducibility, they must ensure the accessibility of each raw sequence by showing accession ID through depositing the raw data in public database.

Authors' response: Thank you for your comments. We modified line 625-629 as follows to ensure the accessibility of each raw sequence: The single-cell RNA sequencing data generated in this paper have been deposited in the NCBI Gene

Expression Omnibus (GEO) database under accession code GSE193279 and in SRA datasets under BioProject PRJNA795657. Raw sequences from each sample are available under accession number PRJNA662018. The raw data from each sample can be accessed using the following login numbers: SAMN24744553, SAMN24744554 and SAMN24744555.

2. Several versions of data analyzing tools were not shown. For examples, the version of Monocles2 in line 546 was not shown. Authors must show versions of all packages they used.

Authors' response: Thank you for your suggestion. In the Methods section, we added the version of each software.

In line 565, we listed the version of Monocle2: Monocle2 (version 2.9.0).

In line 575, we listed the version of velocity.R: velocity.R (version 0.6).

In line 580, we listed the version of Gene-switch: Gene-switch (version 0.1.0).

3. About Gene-switch analysis (from line 558), they did not show any references or URLs. If several URL or references do not exist, they must upload all of codes they perform in public repository (e.g. Github).

Authors' response: Thank you for the kind reminder. In the revised manuscript, we have added the version and reference in the method section (line 580). Codes have been uploaded to Github (https://github.com/superly917/silk_gland_atlas) and Code availability section has been added in the resubmitted manuscript. R Scripts for QC, clustering in Seurat (version 3.1.1) and Monocle2 (version 2.9.0) used for pseudotime analysis were all provided in Supplementary Software 1. Software's license for use in this manuscript is Apache License 2.0.

4. Detailed conditions of qRT-PCR reactions were not shown. (In line 570-577).

Authors' response: Thank you for bringing this issue to our attention. The detailed experimental information for qRT-PCR has been updated in the resubmitted version (line 594-598).

5. The resolution of fig.5 is very low. Replace them to one with high resolutions.

Authors' response: We apologize for the unclear readability caused by the low resolution of Fig. 5. In the revised manuscript, we have updated this figure to meet the resolution requirements (Some representative marker genes in the original Fig. 5 have been separately placed in the newly added Supplementary Figure 6).

6. They used several gene IDs in this manuscript. However, detailed information of these gene set data they used was not shown. Which data? Version and accessing points must be shown in method section.

Authors' response: Thank you for this kind suggestion. Detailed information, including gene ID, gene symbol, gene ID of silkbase and description, TF family, GO terms, and pathways of all genes in the manuscript, has been summarized in Supplementary Data 16 and updated in the Methods section to allow the reader to more conveniently access the gene information.

7. For more clearly understanding the author's works, I strongly recommend to add Scheme diagram of data analysis in the end of introduction. Also, summary diagram of the manuscript must be added in discussion part. The readers must easily understand your works.

Authors' response: Thank you for this valuable suggestion. According to your suggestion, we carefully revised the data analysis process and the highlights of the full manuscript. A schematic diagram of data analysis and summary diagram have been provided in the Introduction section and Discussion section, respectively. We hope that our data analysis and highlighted results are now presented at a glance.

Minor points

1. Fig 6b is poor recognizable because of complicated colors. Change it to dash style or other expression style.

Authors' response: We apologize for the unclear figure. In the resubmitted version, we have updated Fig. 6b to make it more concise and easy to identify.

2. Fig 5, alphabetical orders in figure and main text are incorrect.

Authors' response: Thank you for bringing this issue to our attention. We have carefully reviewed the manuscript and revised these errors.

In addition, we have also made some revisions to the text, figures, methods, references, and corresponding data, which are presented in the resubmitted version. Again, thank you for your kind comments and suggestions.

Reviewer #2:

In this manuscript, Ma and colleagues attempted to identify the cellular atlas of silk glands in *Bombyx mori* and to elucidate the molecular mechanisms involved in silk protein production at the single-cell level by single-cell sequencing. Overall, this manuscript is novel and interesting in terms of experimental purpose and experimental technique. However, there are still some problems in this manuscript in terms of bioinformatics analysis and description of results. Authors need to make substantial revisions to be considered for publication.

Authors' response: We are grateful for your comments on our manuscript, which were very helpful for improving our manuscript and provide encouragement for our future research. Based on your comments and suggestions, we tried our best to improve the manuscript in terms of the bioinformatics analysis and description of the results. Our point-by-point responses are presented below.

1. The samples selected by the authors is not representative. As described by the author, 'the majority of protein synthesis begins on day 3 of the fifth instar'. The author claims that there is no way to take this sample due to technical problems. In fact, the authors can achieve this time point analysis using single-nucleus techniques. The lack of samples at this time point could have important implications for the description of the results of this study. Although I don't know much about the silkworm and the silk spinning mechanism, it is obvious that it is quite difficult to reveal the molecular mechanism of silk protein production from the results obtained by the author at the time point of the sample.

Authors' response: Thank you for this comment and suggestion. We agree with you that Day 3 of the fifth instar larvae (abbreviated as 5L3D) is the beginning of mass synthesis of silk proteins (the attached figure shows a detailed overview of silk gland development). In our previous studies, we performed related research on the regulation of silk protein synthesis and secretion in the late larval stage and found that silk protein is also synthesized at the early larval stage. However, silk gland development and silk protein synthesis in embryonic and early larval stages remain largely unknown because of the difficulty of obtaining a sufficient amount of the silk gland at each time point and immature technical methods. In this paper, we aimed to advance the understanding of this issue by conducting single-cell sequencing. Therefore, we selected three representative samples with characteristics, including 8 days post-egg laying (E8D) stage, which represents the embryonic stage after the complete formation of the silk

gland; 1L1D stage, the first day of larvae, at which the silkworm secretes silk with excellent mechanical properties; and the first larval molting (1LM) stage, a period in which silkworm larvae undergo the first organ renewal. We believe that the analysis of these samples would facilitate the study of silk glands to the resolution of single cells.

In addition, the efficient synthesis of silk protein will still be one of the focuses of our research in the future. According to the characteristics of silk protein synthesis in the late larval stage, research on the protein level has attracted increasing attention. At present, because single-cell nuclear sequencing technology only captures relevant gene expression in the nucleus and information about the synthesis of proteins in the cytoplasm may be lost, we have not yet adopted this method. We look forward to using new technologies to conduct in-depth research on silk gland development and silk protein synthesis in the late larval stage to obtain a new perspective.

2. The author simply divided silkworm silk gland cells into 10 clusters based on bioinformatics. In fact, as long as we adjust the parameters, the number of cell clusters will change, and the entire analysis results will change. Why did the authors not identify silk gland cells? I think the authors should try to identify SG cell types. Of course, it is difficult for non-model organisms, but the work still needs to be done. Clusters would be much easier to follow using cell type names.

Authors' response: Thank you for this professional suggestion. We absolutely agree with your viewpoint that identification of cell types would enable a much easier understanding of the biological characteristics. This study is the first to identify and classify SG cells at the single-cell level, and most of the identified marker genes are novel genes. Based on your suggestion, we performed a detailed analysis according to the annotation of marker genes and the enriched GO and KEGG terms and have provided the name of each cluster. These results have been updated in the revised manuscript (please see the section “Identification of cell types in the SG” for detailed

description). Of course, the naming principle is based on the results of the bioinformatics analysis, and the function of many marker genes requires further experimental verification, which will help to further reveal the cell types in the SG.

Thank you again for this valuable suggestion.

3. The results of the trajectory of SG development accounts for a large proportion of the full manuscript. However, the authors do not specify which cells are the real starting point. In fact, if the authors did not identify the SG clusters, it would be impossible to know which cell types belonged to the early differentiated cells. If the exact starting point of differentiation is not specified in pseudotime analysis, the subsequent analysis results are less credible.

Authors' response: Thank you for your comments. In our study, we determined the starting point based on the actual developmental order. According to the results of our pseudotime series analysis shown in Fig. 4, the three pseudotime series have a common feature, namely, the earliest E8D samples appear on the left of the pseudotime series. With the change in pseudotime, the cells are well arranged into E8D, 1L1D and 1LM. Therefore, from a biological perspective, we believe that the designation of cell types, including LSFs (present in the ASG), SPSs and ERSSs (present in the MSG) and FPSs (present in the PSG), as the starting point of pseudotime is a reasonable approach.

4. In this manuscript, the authors use Immunostaining to detect marker genes. The antibodies used by the authors are polyclonal antibodies from Zoonbio Biotechnology (Supplementary Table 2). The effect of this polyclonal antibody seems to work well. Is this result reliable and will there be false positives? Is it a commercial antibody or a self-made one? The author needs to mark clearly in the text. And the country and region to which the company belongs also needs to be indicated. My concern is that generally everyone uses Fluorescent in situ hybridization (FISH) or in situ hybridization (ISH) to validate marker genes. Why don't the authors use this method?

Authors' response: Thank you for this question. The polyclonal antibodies used in our manuscript are commercial antibodies produced by Zoonbio Biotechnology Co., Ltd., which is located in Nanjing, Jiangsu Province, China (homepage: <http://zoonbio.bioon.com.cn/>)(deaitailed please see Supplementary Table 2). In our manuscript, we used IgG as a negative control and established a division of labor in terms of silk gland anatomy, immunofluorescence experiments and data analysis to ensure the objectivity of the results. In addition, our group has extensive experience in

studying the silk gland. Some silk fibroin and sericin antibodies we used have been shared with other research groups and proven to be effective. The corresponding results have been updated in Supplementary Fig. 2 in the revised version.

As you suggested, fluorescent in situ hybridization (FISH) and in situ hybridization (ISH) are two very important and effective experimental methods. However, because the silk gland is a highly specialized silk-spinning organ and its most important biological function is to synthesize and secrete silk protein, most of the previous studies on silk glands still use immunofluorescence staining for localization. This method detects the protein level using the high specificity between antigen and antibody, which has the characteristics of high sensitivity and fast speed. In addition, immunofluorescence staining has been widely used in single-cell studies, such as *Drosophila* wing disc, ovary and human corneal limbus. FISH and ISH are mainly used to detect nucleic acid levels. Therefore, we comprehensively considered and chose the immunofluorescence localization method, and the experimental results are very conclusive.

Minor Points

1. The author emphasizes the use of 2000 intact SGs for each sample. How do you ensure that such a large number of samples are collected in a short period of time while still maintaining their validity.

Authors' response: Thank you for this question. The dissection of a large number of complete silk glands from individuals at embryonic and early larval stages is very difficult. As shown in Fig. 1b, the silk glands are very small and must be dissected under an anatomical microscope. Although our research group has long been committed to researching the silk gland and has extensive experience in dissecting silk glands at various developmental stages, the dissection of a large number of embryonic and early larval silk glands in the shortest possible time is indeed a very challenging and pioneering attempt for our group. Therefore, the authors of this paper have practiced for several months to dissect the silk glands from E8D, 1L1D, and 1LM individuals to ensure the proficiency and speed of operation. In addition, the instruments and equipment used for sampling and the experimental environment were optimized to ensure the integrity and efficiency of sampling. We also used a programmed cryopreservation container to preserve the samples in batches as a method to increase the cell viability in the samples. The samples were subjected to follow-up experiments using the 10 × Genomics platform within half an hour after dissociation. Through the rigorous, meticulous and efficient operation of each step, the reliability of the final data is ultimately ensured.

2. Which reference genome did the authors use? The authors did not indicate in the Methods section (line 521). Two genome IDs appear in the text. It's puzzling. Line 126, BGIBMGA011721 and others LOC. LOC seems to be from NCBI (Bmori_2016v1.0 ?), why didn't the authors use the latest published genome of the silkworm as the reference genome (doi: 10.1016/j.ibmb.2019.02.002 OR doi: 10.1093/nar/gkz919)?

Authors' response: Thank you for bringing this issue to our attention. We have added the details of the reference genome to the revised version. We chose the classic version of [NCBI](https://ftp.ncbi.nlm.nih.gov/genomes/all/GCF/000/151/625/GCF_000151625.1_ASM15162v1/GCF_000151625.1_ASM15162v1_genomic.fna.gz) (URL: https://ftp.ncbi.nlm.nih.gov/genomes/all/GCF/000/151/625/GCF_000151625.1_ASM15162v1/GCF_000151625.1_ASM15162v1_genomic.fna.gz) and used the corresponding mitochondrial data for quality control of single-cell data. The reason why we chose this version of the reference genome is that many previous omics studies on the regulation of silk gland development and silk protein synthesis are based on this version (gene ID starts with LOC), which is widely used. The single-cell transcriptional atlas of the silk gland analyzed using this version will be better compared with previous studies. In addition, many previous studies on transgene and gene editing conducted by a lot of groups including our group were designed based on this version of the reference genome, which has shown to be one of the high-quality versions. Therefore, we selected this version of the reference genome in this single-cell sequencing analysis for better research continuity. BGIBMGA011721 (GenBank accession number: NM-001173285.1) in line 116 is a cuticular protein gene specifically expressed in the ASG.

3. In figure 1d, the UMAP is not as clear as in other articles. The UMAP results in this manuscript look a little out of the ordinary. Although the author removes likely multiplet cells by some methods in the text. Has the author considered further removal of polycells through tools such as DoubleFinder?

Authors' response: Thank you for this comment. For low-quality and multiplet cells, the corresponding UMI/gene will be very low or very high, respectively. Therefore, we removed possible low-quality and multiplet cells according to the Gaussian distribution characteristics of the UMI/gene of each sample. In addition, the number of captured cells in our three samples ranged from 5351 to 7513. According to the corresponding table of recovered cells and multiplet cell rates provided by 10× Genomics below, the multiplet cell rate of our three samples was maintained at approximately 5%, and the three samples contained only a small number of multiplet cells (please see Table 1 below).

Table 1. Corresponding number of recovered cells and multiplet cell rates

Multiplet Rate (%)	# of Cells Loaded	# of Cells Recovered
~0.4%	~870	~500
~0.8%	~1700	~1000
~1.6%	~3500	~2000
~2.3%	~5300	~3000
~3.1%	~7000	~4000
~3.9%	~8700	~5000
~4.6%	~10500	~6000
~5.4%	~12200	~7000
~6.1%	~14000	~8000
~6.9%	~15700	~9000
~7.6%	~17400	~10000

In addition, we performed a DoubleFinder test on E8D samples. The possible multiplet cells in DoubleFinder were compared with the cells filtered out by our QC procedure. Most of the possible multiplet cells in the DoubleFinder results were filtered out by our current QC results. The results (a ZIP folder named 'Results of Doublefinder analysis' in the resubmitted version) are provided with the response letter.

4. Line 186 and 187, Authors should only select insect libraries for KEGG analysis, not vertebrate data. These results are clearly contrary to common sense and do not deserve further elaboration.

Authors' response: Thank you for bringing this issue to our attention. According to previous studies, the silkworm genes are similar to those of higher mammals, such as humans. More than 3484 silkworm genes are homologous with those of vertebrates, which is a very important discovery of the silkworm genome project (Xia, Q. & Xiang, Z. *the genome of silkworm [M]*, Science Press, 2013). Compared with *Drosophila*, *B. mori* has a relatively large genome and appropriate complexity. It is very similar to humans in terms of basic life systems, material metabolism, energy metabolism and heredity. Interestingly, *B. mori* has similar disease models with human tumors, diabetes, abnormal nerve conduction, abnormal behavior, infertility, and developmental

malformations. Based on these important foundations, the KEGG pathway of vertebrates is usually covered in the enrichment analysis of silkworm, which is why we retained these terms in the single-cell data analysis.

5. Line 191-192, I think the 'above results' are not accurate to demonstrate the high heterogeneity of SGs. It's normal for a single organ to be made up of many kinds of cells. Heterogeneity requires objects of comparison.

Authors' response: Thank you for this suggestion. We have updated the detailed analysis and description of the definition and comparison of the heterogeneity of cell types in the resubmitted version.

6. Line 511, Why did the authors not choose the silkworm strain DAzao, whose genome has already been published.

Authors' response: Thank you for this question. The silkworm strains *Dazao* and *Nistari* are commonly used materials for silkworm research and have a highly consistent genetic background. The main difference between them is that *Nistari* is a nondiapause strain while *Dazao* is a diapause strain (To our knowledge, *Dazao* is actually derived from a nondiapaused strain). More importantly, many previous studies have confirmed that cloned gene sequences are highly consistent, regardless of which strain is used. In addition, the silkworm strain *Nistari* is not in diapause and can be raised continuously many times a year, which is very convenient for material acquisition and is widely used in current research.

7. Many conclusions in the text have overstate. Authors need to pay attention to accurate description of results.

Authors' response: Thank you for your valuable comments. In the resubmitted version, we have tried our best to improve the scientific value of the manuscript by performing further analyses of the scRNA-seq data, increasing the accuracy of the description of the results, and discussing the results in greater depth.

8. The resolution of some pictures needs to be improved, as shown in Figure 5b, d, f.

Authors' response: We apologize for the unclear readability caused by the low resolution of the picture. In the revised manuscript, we updated a lot of figures including the Figure 5 and hope that the updated figures could meet the resolution requirements.

Once again, on behalf of all the authors, I would like to thank you for your comments and valuable suggestions.

Reviewer #3:

In this study, the authors conducted single-cell transcriptomic analysis on the silkworm silk gland of three early developmental stages. Generally, this study provided new data and added our understanding of silk gland development at early developmental stages, which is of potential value. However, I had expected much more than what the manuscript told at this step.

- First of all, the whole manuscript is too much descriptive, with so many trivial findings but without compelling highlights or emphasis. I can't catch the novel point(s), from the manuscript. For example, As to the result section "Dynamic changes in gene expression in SG cells", I'm really drown in the batch of turn-on and turn-off genes. Similar situation was also in "Regulation of SG growth and development in early stages". What can the author infer from the massive description?

Authors' response: Thank you for this question. According to your suggestion, we have tried our best to improve the scientific value and readability of the manuscript by performing further analyses of the scRNA-seq data, increasing the accuracy of the description of the results, and discussed the results in the resubmitted manuscript. Our study is the first to explore the whole picture of the silk gland at the single-cell level. We propose that these data will be very helpful for further studies of the regulation of silk gland development and silk protein synthesis at a higher resolution. Therefore, we want to provide as much useful information as possible in this paper.

Thank you again for your kind reminder. We also realize that refining and improving these results are necessary to highlight the results and key points. Therefore, we have carefully refined and revised the full text and provided a schematic diagram of data analysis and a summary diagram in the Introduction section (Fig.1) and Discussion section (Fig.8), respectively, to ensure that the presentation is as bright and clear as possible. Of course, our research conducted at the single-cell level of the silk gland opens a door. Many unknown and interesting biologically significant factors underlie the data, which are worthy of long-term exploration. We look forward to communicating with you and receiving your guidance, which will be a great honor and help for us.

- Furthermore, if "mitotic division of SG cells ceases after stage 25 (6 days post-egg laying)¹², which means that the SG has completed morphogenesis at this stage", then why new cell types appear or disappear at different developmental stages? I think maybe cell differentiation occurs during silk gland development. If this is the case, collection of more samples with different developmental stages, particularly the last

larval stage when the mature silk gland may have distinctive characters compared with early-stage one. Regrettably there seems no good approach for SC-RNA seq on silk gland at this stage. This issue becomes difficult and hard to be addressed.

Authors' response: Thank you very much for your comments, and we apologize for the confusion. The reason why new cell types appear in the silk gland at different developmental stages is that cells perform different biological functions at each developmental stage, and different marker genes are expressed at high levels in the cell type at the corresponding stage rather than undergoing cell differentiation. As you suggested, the collection of more samples at different developmental stages, particularly the last larval stage when the mature silk gland may have distinct characteristics compared with the early-stage silk gland, is important to reveal the molecular mechanism regulating silk protein synthesis. Therefore, in our previous studies, we performed related research on the regulation of silk protein synthesis and secretion in the late larval stage (as shown in the figure below) and found that silk protein are also synthesized in the early larval stage. However, silk gland development in the embryonic stage and silk protein synthesis in the early larval stages remain largely unknown because of the difficulty of obtaining materials and immature technical methods. In this paper, we aimed to advance the understanding of this issue by conducting single-cell sequencing, which could facilitate the study of silk glands at the resolution of single cells.

In addition, the transcriptional regulation of the silk gland at the single-cell level in different developmental stages will still be one of the focuses of our research in the future. At present, due to technical limitations, the cell size of the silk gland at late larval stages exceeds the requirements of the 10× Genomics platform, and single-cell nuclear sequencing technology only captures relevant information in the nucleus. We look forward to using new technologies (for example, single-cell proteome sequencing) to conduct in-depth research on silk gland development and silk protein synthesis in the late larval stage to obtain a new perspective, and to obtaining more suggestions from you on our future work.

Thank you again for your valuable suggestion.

● Specifically, Results of clustering and annotation of the cell types are not well evidentiary supported. In addition, the demonstration and explanation of the results are confusing.

1. Fig 1g and 1h, of course one can get specific or highly expressed genes in each cluster. The problem is how can we infer from these so-called marker genes? What is their biological significance?

Authors' response: Thank you for bringing this issue to our attention. The top 10 highly expressed marker genes of each cluster are shown in Fig. 2c (Fig. 1g in the first submission), and cluster-specific marker genes are shown in Fig. 2d (Fig. 1h in the first submission), in which we identified many marker genes that are active in specific cell types during SG development, including some genes that are expected based on previous studies. For example, the fibroin protein-coding gene *P25* (C1), some cuticular protein genes that are necessary for ASG development, and *Btl* (C4) may be involved in the formation of lumen and the limitation of SG cell number. The biological significance of these representative known marker genes and some highly expressed unknown marker genes are described in detail in other parts of our manuscript.

2. Line126-134. What's the relationship of these genes with those shown in Fig 1g and 1h? Since they are all marker genes, the authors should not explain them separately. Specifically, the biological significance of the genes labeled by geneID and known markers in different regions of the silk gland should be demonstrated. Why these genes could annotate different region of the silk gland? By the way, the genes are all shown with gene ID but not gene name, hampering our understanding of their biological significance. Results of Immunofluorescence verification did not clearly supported the annotation. The authors need to explain the results.

Authors' response: Thank you for your suggestions, and we apologize for the lack of

clarity. According to your suggestion, we carefully revised the description of Fig. 1g and 1h in the submitted manuscript (Fig. 2c and 2d in the resubmitted version). Detailed information, including the gene ID, gene symbol, description, and TF family, of all marker genes mentioned in the manuscript has been summarized in Supplementary Data 16 and updated in the Methods section to allow the reader to more conveniently understand their biological significance.

In terms of cell type annotation, different silk proteins are secreted in different regions of the silk gland. The ASG does not secrete silk proteins and is simply a processing lumen, and two known ASG-specific cuticle protein genes, *BmASSCP2* and *BGIBMGA011721*, were identified; the MSG secretes sericin proteins, including sericin 1 (Ser1), sericin 2 (Ser2), and sericin 3 (Ser3); and the PSG secretes fibroin proteins, including fibroin H (fibH), fibroin L (fibL), and P25. These genes are classic markers that are useful to annotate cell types, as shown in Fig. 3a. We verified their expression in the ASG, MSG and PSG at three developmental stages using qRT - PCR, which were shown as below.

1) Featureplots of the expression pattern showed that *BmASSCP2* and *BGIBMGA011721* were expressed at high levels in C4 and C9 (Supplementary Fig. 1).

2) qRT - PCR results showed that *LOC101743237* (shared in C4 and C9) was predominantly expressed in the ASG, and the specific marker genes *Btl* (in C9), *LOC101746180* (in C5) and *LOC101740197* (in C8) were predominantly expressed in the ASG (Fig. 3a).

3) Immunofluorescence verification of *LOC101746180* (in C5) and *LOC101740197* (in C8) showed that they were significantly expressed in the ASG of 1LM individuals, which further supported this identification of cell types in the SG (Fig. 3b). Collectively, we conclude that C4, C5, C8 and C9 belong to the ASG.

4) The MSG-specific marker gene *Ser2* was predominantly expressed in C10 (Supplementary Fig. 1).

5) The cluster-specific markers *LOC101745308* (in C3), *LOC101740733* (in C6) and *LOC10174471* (in C10) were specifically or highly expressed in the MSG (Fig. 3a).

6) Immunofluorescence verification showed that *LOC101745308* (in C3) was expressed at significantly higher levels in the MSG (Fig. 3b). Collectively, we conclude that C3, C6 and C10 belong to the MSG.

7) The PSG-specific marker genes *fibH*, *fibL*, and *P25* were all predominantly expressed in C1, C2, and C7. Since *P25* is also a top 10 marker gene of C1, we also verified it using immunofluorescence staining, and the results showed that it was expressed at significantly higher levels in the PSG (Fig. 2b). Two cluster-specific marker

genes, *LOC101746861* (in C1) and *LOC101743535* (in C7), were relatively specifically expressed in PSG (Fig. 2a). Collectively, we conclude that C1, C2 and C7 belong to the PSG.

3. I'm confused in the enrichment analysis. Only top 10 marker genes, how to generated enrichment analysis?

Authors' response: Thank you for bringing this issue to our attention. We conducted the enrichment analysis of all marker genes in each cluster and then selected the top 10 for display when drawing the heatmap. We have revised the statement in the revised manuscript for clarity.

4. Cell clusters specific to certain region of the silk gland, could be better used for sub-cluster analyses.

Authors' response: Thank you for this kind suggestion. We absolutely agree with your viewpoint that subcluster analyses are very important for obtaining a better understanding of the biological functions of marker genes and cellular heterogeneity. According to your suggestion, we performed a subcluster analysis of each cluster, and some interesting findings were added to the manuscript. Thank you again.

Some statements are not precise, even wrong.

- Line 483-485: "...suggest a functional transition of SG cells from the embryonic to larval stage, the first metamorphosis". I think this is a wrong statement. I don't think development from embryonic to larval stage is a process of metamorphosis. Particularly, at E8D, the embryo is factually well developed and similar to newly hatched silkworm

larva. As to results on hormone regulation on silk protein synthesis. As I know, there are many recent advances in this issue. The authors need to discuss the results deeply, consulting recent references rather than merely objective implication. For instance, Line 492-493:” These results strongly suggest that 20E signaling plays a crucial role in inhibiting silk protein synthesis in SG cells during larval molting”.

Authors' response: Thank you for your suggestions. In the resubmitted version, we have carefully revised the manuscript and corrected the inaccurate statements. In particular, the hormone regulation you proposed was further discussed, and the latest references have been added. We hope these efforts increase the scientific significance and readability of the manuscript. Thank you again for your constructive suggestions.

Other:

Description of sampling and sequencing quality is not clear. As I understand, totally three samples was subjected to single-cell RNA-seq, respectively? If this is the case, then how many cells sequenced in each sample? In addition, the reference genome information was not provided. Which version was used for reads mapping? How about the mapping rate?

Authors' response: Thank you for your comments, and we apologize for the lack of clarity. As you noted, three samples (E8D, 1L1D and 1LM) were subjected to single-cell RNA-seq analysis. We emphasize this information in line 68-72, line 83-86, and line 526-534 to more clearly describe the sequencing of samples. The reference genome information was added to the Methods section in line 537-538. According to the results from Cell Ranger, 5351 cells, 6000 cells and 7513 cells were captured in E8D, 1L1D and 1LM samples, respectively, and the corresponding genome mapping rates were 88%, 84.8% and 74.2%, respectively.

- Fig7 The first letter of the text should be capitalized.

Authors' response: Thank you for bringing this issue to our attention. We have modified Fig. 7 according to your suggestions and updated it in the resubmitted version.

Once again, on behalf of all the authors, I would like to thank you for your comments and valuable suggestions. We have tried our best to revise and improve the manuscript according to your suggestions and hope these efforts will provide a better understanding of our results.

REVIEWERS' COMMENTS

Reviewer #1 (Remarks to the Author):

The authors addressed properly my all comments, and I think that the manuscript was improved. I expressed respects for their efforts. I recommended it for publication.

Reviewer #2 (Remarks to the Author):

The author has responded to my concerns.

Reviewer #3 (Remarks to the Author):

I'm happy that the authors responded to my concerns carefully and made a lot of revisions and modifications to the manuscript.

Response to the Reviewers

Reviewer #1:

The authors addressed properly my all comments, and I think that the manuscript was improved. I expressed respects for their efforts. I recommended it for publication.

Authors' response: We deeply appreciate your review on our manuscript. Your valuable comments and suggestions greatly help us to improve the quality of the manuscript and provide great encouragement for our future work. In the process of revising the manuscript, we also recognize more the importance of detailed description of methods, so that readers can more clearly understand all the methods used in this study. We look forward to communicating with you and receiving your guidance, which will be a great honor and help for us.

In the resubmitted version, we have carefully checked and revised our manuscript item by item based on the author checklist to meet the publication requirements of Nature Communications.

Thank you again and best regards.

Reviewer #2:

The author has responded to my concerns.

Authors' response: We are grateful for your comments and kind suggestions on our manuscript, which are valuable and very helpful for improving our paper, as well as the encouragement and important guiding significance to our research work. In the processing of revising, we really realized that clusters would be much easier to follow using cell type names as you suggested. This study is only a starting point for the study of silk glands at the single-cell level and we look forward to using new technologies to conduct in-depth research on silk gland development and silk protein synthesis in the late larval stage to obtain a new perspective.

In the resubmitted version of the manuscript, we have carefully checked and revised our manuscript item by item based on the author checklist and revised portion are marked in yellow. These changes will not influence the content and framework of the paper.

Once again, on behalf of all the authors, I would like to thank you for reviewing our manuscript.

Reviewer #3:

I'm happy that the authors responded to my concerns carefully and made a lot of revisions and modifications to the manuscript.

Authors' response: On behalf of my co-authors, we would like to express our great appreciation to you for your comments and kind suggestions on our manuscript, which will be the encouragement and important guiding significance to our further researches. In the processing of revising, we fully realized that refining some results are necessary to highlight the key points, for example, a schematic diagram of data analysis and a respective summary diagram in the Introduction section (Fig.1) and Discussion section (Fig.8) make the presentation brighter and clearer and improve the readability of our manuscript. In addition, the transcriptional regulation of the silk gland at the single-cell level in different developmental stages will still be one of the focuses of our research in the future. We look forward to using new technologies (for example, single-cell proteome sequencing) to conduct in-depth research on silk gland development and silk protein synthesis in the late larval stage to obtain a new perspective. We look forward to communicating with you and receiving your guidance, which will be a great honor and help for us.

Thank you again and best regards.